# Application of TGA/c-DTA for Distinguishing between Two Forms of Naproxen in Pharmaceutical Preparations

**DOI:** 10.3390/pharmaceutics15061689

**Published:** 2023-06-08

**Authors:** Paweł Ramos, Barbara Klaudia Raczak, Daniele Silvestri, Stanisław Wacławek

**Affiliations:** 1Department of Biophysics, Faculty of Pharmaceutical Sciences in Sosnowiec, Medical University of Silesia in Katowice, Jedności 8, 41-200 Sosnowiec, Poland; 2Faculty of Mechatronics, Informatics and Interdisciplinary Studies, Technical University of Liberec, 461 17 Liberec, Czech Republic; barbara.klaudia.raczak@tul.cz (B.K.R.); stanislaw.waclawek@tul.cz (S.W.); 3Institute for Nanomaterials, Advanced Technologies and Innovation, Technical University of Liberec, Stdentská 2, 460 01 Liberec, Czech Republic; daniele.silvestri@tul.cz

**Keywords:** naproxen acid, naproxen sodium salt, pharmaceutical preparation, TGA, c-DTA, UV spectrophotometry, HPLC, FTIR, colorimetric analysis

## Abstract

Naproxen is one of the most used non-steroidal anti-inflammatory drugs (NSAIDs). It is used to treat pain of various origins, inflammation and fever. Pharmaceutical preparations containing naproxen are available with prescription and over-the-counter (OTC). Naproxen in pharmaceutical preparations is used in the form of acid and sodium salt. From the point of view of pharmaceutical analysis, it is crucial to distinguish between these two forms of drugs. There are many costly and laborious methods to do this. Therefore, new, faster, cheaper and, at the same time, simple-to-perform identification methods are sought. In the conducted studies, thermal methods such as thermogravimetry (TGA) supported by calculated differential thermal analysis (c-DTA) were proposed to identify the type of naproxen in commercially available pharmaceutical preparations. In addition, the thermal methods used were compared with pharmacopoeial methods for the identification of compounds, such as high-performance liquid chromatography (HPLC), Fourier-transform infrared spectroscopy (FTIR), UV-Vis spectrophotometry, and a simple colorimetric analyses. In addition, using nabumetone, a close structural analog of naproxen, the specificity of the TGA and c-DTA methods was assessed. Studies have shown that the thermal analyses used are effective and selective in distinguishing the form of naproxen in pharmaceutical preparations. This indicates the potential possibility of using TGA supported by c-DTA as an alternative method.

## 1. Introduction

Naproxen (NA) belongs to the family of non-steroidal anti-inflammatory drugs (NSAIDs) of phenylpropionic acid derivatives. NA has the strongest analgesic, anti-inflammatory, and antipyretic effect among drugs belonging to phenylpropionic acid derivatives [1,2,3,4]. The mechanism of action of naproxen is mainly based on the inhibition of isoform one cyclooxygenase (COX-1) [2,4,5]. For this reason, it causes ulcerative damage to the stomach and gastrointestinal disturbances typical of NSAIDs. Naproxen crosses the blood–placental barrier and is excreted in breast milk. NA has a long half-life of up to 15 h, and peak blood levels are reached after oral administration in 2 to 4 h [1,2,4]. Naproxen has been used in the treatment of inflammation, mainly in rheumatoid arthritis and arthrosis. Ninety percent of Naproxen excretion is performed by the kidneys, with 70% mainly unchanged. For this reason, it cannot be used in people with kidney damage due to the possibility of its accumulation [2,3,4]. Naproxen comes in tablets, suppositories, and dermal preparations for external use. The tablets may contain the acid form of naproxen or be in the form of sodium salt, which is more soluble and is absorbed faster (maximum concentration in the blood after 1 h) (Figure 1) [1,2,5]. Naproxen is a weak acid with p*K*a = 4.15, which determines the rate of its absorption [6,7]. On the other hand, it is known that weak acids in the form of salts dissolve faster in an aqueous environment; therefore, naproxen in sodium form is more rapidly dissolved in the environment of body fluids and more rapidly absorbed into the plasma [6,7,8]. This allows for a faster analgesic effect of naproxen in the form of salt compared to naproxen acid, which has been confirmed in clinical trials [8]. As a result, most new naproxen preparations contain sodium salt due to the faster effect of action [2,5].

The ability to identify the form of naproxen in pharmaceutical preparations is important due to its use in the forms of acid and sodium salt. The ability to distinguish between naproxen preparations is essential from the point of view of quality control, determination of active pharmaceutical ingredient (API) purity, possible impurities, and identification of counterfeit drugs.

In the pharmaceutical industry, many methods are used to identify the type of API [9,10,11,12]. Advanced methods such as NMR, HPLC, FTIR, XRPD, or Raman spectroscopy are accurate methods, but often the methodology and performance of determinations are time consuming, complicated, and require expensive equipment [13,14,15,16,17]. In turn, more cost-effective and simpler methods such as UV-Vis spectrophotometry, microscopic observations, or colorimetric analyses do not always give the expected results. Therefore, methods are constantly being sought that will be selective and at the same time simple and inexpensive.

An example of such methods are thermal analyses such as thermogravimetry (TGA) or differential thermal analysis (DTA). These methods have shown their usefulness in our previous studies for the identification of single- and dual-component preparations containing theophylline [18]. In addition, other authors have demonstrated their usefulness in the pharmaceutical industry for the identification of polymorphic substances [19,20], the influence of ambient relative humidity on vapor sorption [21], thermal stability [22,23], or compatibility of excipients with API [24,25,26,27] and API with API [28], as well as quality control in pharmacies [29,30,31].

The aim of this study was to use thermal analysis (TGA and c-DTA) to evaluate the difference in the thermal decomposition profiles of naproxen acid and sodium salt. These differences provide the basis for identifying the type of naproxen in commercial pharmaceutical preparations. In order to determine the sensitivity and selectivity of the thermal methods, the obtained results were compared with a preparation containing nabumetone, a close structural analogue of naproxen used as an NSAID. In addition, the work compares thermal analyses with other techniques used in pharmacy to identify drugs, i.e., UV-Vis spectrophotometry, HPLC, FTIR, and simpler methods such as colorimetric analysis.

## 2. Materials and Methods

### 2.1. Tested Samples

This work tested two forms of naproxen in acid (NA) and sodium salt (NS). Pure NA and NS were purchased from the Sigma-Aldrich Company and used as standards. The measurements were performed for commercial pharmaceutical preparations containing naproxen acid and sodium salt. Information on the tested drugs is given in Table 1. In addition, nabumetone in the form of a pharmaceutical preparation Nabuton VP (ICN Polfa Rzeszów S.A., Rzeszów, Poland) in a dose of 500 mg was used in the work.

### 2.2. TGA and c-DTA Measurements

The thermograms of naproxen acid, naproxen sodium salt and pharmaceutical preparations containing NA and NS were determined by thermogravimetric analysis. The thermogravimeter TG 209 F3 Tarsus produced by Netzsch (Selb, Germany) was used. For tested samples, thermogravimetric dynamic measurements were made. Additionally, to improve the readability of the TG curves, the first (DTG) and second (D2TG) derivative TG curves were simultaneously measured [32].

Calculated differential thermal analysis (c-DTA) was performed to determine exothermic and endothermic events [33]. In these measurements, multiple-point temperature calibration was carried out using c-DTA. For this method, the beginning temperatures of the melting peaks of high-purity reference materials (In, Sn, Zn, Al, BaCO_3_, and Au) over the entire temperature range were determined.

For all tested samples, thermal measurements were made under the same conditions, i.e.,
sample weight: 10 mgheating rate: 10 L/mintemperature range: 35 °C to 600 °Catmosphere under measurement: nitrogen (N_2_)total flow nitrogen rate: 40 mL/min

For thermal measurements, the tested samples in the form of powder were placed in a Al_2_O_3_ crucible type. The weight of the samples was determined using the CPA analytical balance (Sartorius, Göttingen, Germany).

All obtained thermal curves were analyzed using Proteus 8.0 software produced by Netzsch company (Germany).

### 2.3. FTIR Methodology

Fourier-transform infrared spectroscopy spectra (4 cm^−1^ resolution at 4000–500 cm^−1^) were obtained with a germanium ATR crystal (NICOLET IZ10, Thermo Scientific, Waltham, MA, USA) with a horizontal ATR attachment with a single reflection angle of 45°. Samples were analyzed as KBr pellets containing the drug.

### 2.4. Density Functional Theory Calculations

The structure of the naproxen acid and sodium salt was constructed in Avogadro software [34] and further optimized using the Orca program package [35] at the B3LYP/def2-TZVP level of density functional theory. The frequencies were obtained at the same level of theory and further processed by Multiwfn software [36]. The results extracted from Multiwfn software were plotted together with experimentally obtained FTIR spectra.

### 2.5. UV-Vis Spectrophotometry

In this work, UV absorbance spectra of the naproxen acid, naproxen sodium salt, and pharmaceutical preparations containing NAS (NA1-NA4) and NSS (NS1-NS5) as the active pharmaceutical ingredients were registered. For this purpose, 10 mg of each of the samples were dissolved in 100 mL of 96% methyl alcohol (Sigma-Aldrich Company, St. Louis, MO, USA). The samples were then mixed and poured into a quartz cuvette placed in a UV-Vis spectrophotometer. UV absorbance spectra were obtained in the wavelength range from 245 nm to 345 nm. The UV-Vis spectrophotometer Thermo Genesys 10S produced by Thermo Scientific (Waltham, MA, USA) was used. To obtain and analyze UV spectra, VisionLite Software (Thermo Scientific Company, Waltham, MA, USA) and Origin 2016 (OriginLab Company, Northampton, MA, USA) were used.

### 2.6. HPLC Methodology

The high-performance liquid chromatography (HPLC) apparatus (UltiMate 3000, Thermo Fisher Scientific, Brno, Czech Republic) with a UV-VIS (VWD-3100) detector was used to separate the naproxen acid and naproxen sodium salt. The flow rate was adjusted to 1200 μL/min. The mobile phases consisted of mixtures of 40% acetonitrile and 60% 0.01 M orthophosphoric acid water solution. The separation of naproxen sodium salt and naproxen acid were conducted in 40 °C with HPLC column type C18 (150 mm × 4.6 mm, 2.6 μm; Phenomenex, Praha, Czech Republic). The injection of the volume was 20 μL. The retention time for naproxen acid was obtained in 4.70 min, and for naproxen sodium salt in 4.0 min. The wavelength or measurement was 230 nm, and the limit of quantification was 0.5 mg/L for both (naproxen acid and naproxen sodium salt).

### 2.7. CIE L*a*b* Measurements and Statistical Analysis

Colorimetric analysis in the 3D CIE L*a*b* color system was performed. For this purpose, the NH 310 colorimeter from 3nh Company (Guangzhou, China) was used. Analyses of color parameters (L*, a*, b*) was done for pure naproxen standards and powdered pharmaceutical preparations.

Parameter L* refers to the lightness of the tested samples. Parameter lightness ranges from 0 to 100, where 0 indicates the ideal black color and 100 indicates the ideal white color. Values obtained from 1 to 99 indicate shades of grey. Parameter a* refers to the redness of the samples. A negative value of redness indicates green, while a positive value of redness indicates red. Parameter b* refers to the yellowness of the samples. Positive and negative b* indicates yellow and blue color, respectively [21,37].

All measurements were done eight times for each sample. The received values were averaged (±SD). A one-way ANOVA test was used to assess statistical significance. The statistical significance was assumed to be *p* < 0.05. Statistical analysis was done using the Statistical software produced by TIBCO Software Inc. (Palo Alto, CA, USA).

## 3. Results and Discussion

### 3.1. Thermal Analysis

In the study, thermal analyses, including TGA, DTG, D2TG, and c-DTA, were used to assess the naproxen type in selected pharmaceutical products. The same measurement conditions were used for the thermal analysis of the tested substances. The use of stable conditions allows for the comparison of the obtained results. Naproxen is used in various forms to produce pharmaceutical formulations [2,5]. Therefore, high-purity standards of naproxen acid and sodium salt were used in the studies.

#### 3.1.1. TGA and c-DTA Measurements of Naproxen Acid

TGA curves and their parameters for naproxen acid standard (NAS) and pharmaceutical preparations containing naproxen acid are presented in Figure 2 and Table 2. The thermogravimetry curve obtained for the naproxen acid standard showed that the thermal decomposition began at 255.0 °C and contained two stages. The total mass loss for both stages was −89.33% (Table 2).

Thermogravimetry curves registered for all tested pharmaceutical preparations containing naproxen acid (NA1–NA4) have the same course and shape as NAS (Figure 2, Table 2). The onset of the decomposition of the tested pharmaceutical preparations is in the range of 257.5 °C to 259.3 °C. These values are practically identical to the used standard. This indicates that the pharmaceutical preparations contain the same API, i.e., naproxen in the form of acid. The tablet mass was pulverized so that the consistency of the tested pharmaceutical preparations was the same as the standard. This is important because the size of the particles has been shown to influence thermal behavior. As the research has shown, the decomposition onset temperature is lower for smaller particles [38,39].

Simultaneously, the first and second derivatives were recorded for TG measurement. This record was made to improve the readability of thermal events of the analyzed samples. Figure 3 and Table 3 show the DTG and D2TG curves and their parameters for the naproxen acid standard and tested pharmaceutical preparations.

NAS has two decomposition stages on the DTG curve (Figure 3a and Appendix A) corresponding with the TG curve (Figure 2 and Appendix A). The major mass loss occurs in the first stage, with a maximum mass loss per minute of −17.96% at a temperature of 290 °C [22]. The second mass loss is smaller compared to the first and occurs at a temperature of 356 °C (−0.93%/min). The second stage is associated with further degradation of naproxen. Analyzing the D2TG curve for pure naproxen, we can observe that the first stage is more pronounced and occurs in the range from 271 °C to 304 °C. The second stage, on the other hand, is less pronounced and occurs in the range of 341 °C to 368 °C. This is due to the fact that most of the compound is degraded in the first stage (Figure 3b, Table 3) [19,22,40].

Analyzing the DTG and D2TG curves of the drugs containing naproxen acid (NA1–NA4), we can observe very similar shapes and events. For the tested pharmaceutical preparations, we can observe a slight shift in the maximum mass loss for the second stage of decomposition (Figure 3 and Appendix A, Table 3). This is likely related to the content of excipients for which maximum mass loss is observed at higher temperatures (e.g., magnesium stearate: 370 °C; croscarmellose: 360 °C; methylcellulose: 358 °C) [26,27,41,42,43,44].

In order to identify endo- and exothermic events occurring during thermal decomposition, calculated differential thermal analysis (c-DTA) measurements were carried out simultaneously for the tested samples.

Figure 4 and Table 4 present c-DTA curves and parameters of pure naproxen acid and pharmaceutical preparations containing NAS as API. For pure naproxen acid, we can observe two peaks on the c-DTA curve. The first peak, with a maximum of 158.7 °C, is endothermic and is related to the melting point of naproxen acid [19,33,41]. The second peak at 314.5 °C is exothermic and associated with the decomposition of naproxen [33]. This peak corresponds with the thermogravimetry curve registered for the naproxen standard.

All c-DTA curve events registered for pharmaceutical preparations containing naproxen acid are coincident with the events for the standard (Figure 4, Table 4). This indicates the possibility of using c-DTA as a method to confirm the results obtained using thermograms. An essential parameter confirming the identity and purity of API in preparations is the melting point (marked with a dashed pink line in Figure 4) [18,33]. The melting point for the tested pharmaceutical preparations had a maximum peak in the range of 155.3 °C to 156.7 °C (Table 4). These values almost entirely coincide with the standard. Slight shifts of the melting point towards lower temperatures are allowed for mixtures of API with excipients [27,33]. The second stage recorded for the tested finished drugs was exothermic, in line with the standard. The maximum second-stage exothermic peak for the pharmaceutical preparations was in the range of 318.7 °C to 323.2 °C (Table 4).

#### 3.1.2. TGA and c-DTA Measurements of Naproxen Sodium Salt

The thermogravimetry curves of the naproxen sodium salt standard (NSS) and pharmaceutical preparations containing naproxen sodium salt in the formulation are shown in Figure 5. The TG curve indicates that the main thermal decomposition of NSS occurs at 375.6 °C. Compared to the decomposition of pure naproxen acid (255.0 °C), naproxen sodium salt is more thermally stable (Table 2 and Table 5) [40]. The total mass loss in all steps was 32.20%.

The TG curves for pharmaceutical preparations (NS1–NS5) containing NSS are more complex than for drugs containing naproxen acid (Figure 2 and Figure 5). This is due to the higher content of excipients in the composition of the tablets. However, by analyzing the decomposition onset temperature of the main mass loss associated with API degradation in the formulation, we can demonstrate the similarity of the NS1–NS5 curves to the standard. The onset of decomposition, accompanied by the most significant mass loss, ranges from 369.5 °C to 375.6 °C. These values are very close to the naproxen sodium salt standard (Table 5).

The first and second TG derivatives were measured simultaneously for a clear comparison of all mass loss stages (Figure 6). The DTG trace showed a more complex degradation for the salt than the naproxen acid [40]. Naproxen sodium salt undergoes four-step decomposition (Appendix A). The first mass loss stage occurred in the temperature range of 65 °C to 76.9 °C, with a maximum peak of 70.6 °C and mass change of −0.89%/min. (Figure 6 and Appendix A, Table 6). This stage is associated with water loss [23,45]. The second stage occurred in the temperature range of 391.7 °C to 409.2 °C, with a maximum peak of 396.7 °C. This stage has the most significant mass loss per minute of −12.23%. The third stage begins at 415.4 °C and is a continuation of the second stage, which can be seen by overlapping the second and third stages into one large peak with two maxima. In the third stage, the mass loss is −7.47%/min. The last stage is onset at 459.4 °C, with maximum mass loss at 462.6 °C (1.73%/min.). The second, third, and fourth stages were related to the decomposition of naproxen sodium salt [40]. As is known, naproxen sodium can be in the form of an anhydrate, a monohydrate, two dihydrate polymorphs, and a tetrahydrate [46]. The formation of pseudo-polymorphic forms may affect the recording of thermal curves. Rajada D et al. showed in their studies that anhydrous sodium naproxen is converted to one of the dihydrate polymorphs already at 25 °C. On the other hand, at a higher temperature (50 °C), anhydrous sodium naproxen is gradually converted into a monohydrate and then into another dihydrate polymorph. The dihydrate polymorphs can transform into tetrahydrates and monohydrates [46].

The DTG and D2TG curves for all pharmaceutical preparations containing naproxen in salt form correlate with the NSS (Figure 6a and Appendix A, Table 6). This result confirms that the tested drugs contain naproxen sodium salt, as indicated by the recorded characteristic peaks on the DTG curves—especially peaks related to the decomposition of naproxen sodium (peaks two to four). The first stage associated with water loss was recorded for the study drugs NS1–NS5 (Figure 5, Table 6). The maximum mass loss for stage one for drugs NS1–NS3 and NS5 was shifted slightly towards higher temperatures and ranged from 80 °C (NS1) to 88.4 °C (NS2). For sample NS4, this stage was the smallest and occurred at the temperature closest to the standard (Table 6). This is likely related to the addition of excipients not present in other formulations. The peaks associated with the NSS decomposition recorded for pharmaceutical preparations fully coincided with the naproxen sodium standard. For tested drugs, the maximum mass loss for the second stage occurred in the range of 388.6 °C (NS4) to 397 °C (NS1); for the third stage, in the range of 411.3 (NS2) to 423.5 (NS4); and for the last stage, in the range of 457 °C (NS5) to 466 °C (NS4).

All tested pharmaceutical preparations (NS1–NS5) contain additional peaks in the DTG and D2TG curves derived from the excipients used (Appendix A, Table 6). The first additional peak for samples NS1–NS3 and NS5 has a maximum mass loss from 48.8 °C (NS3) to 57.2 °C (NS2) and is derived from microcrystalline cellulose. Microcrystalline cellulose exhibits a maximum mass loss at 46.4 °C, related to the release of water [26,47]. Another excipient-derived peak in the DTG curve was recorded only for the NS2 and NS4 samples, with a maximum mass loss at 172.3 °C and 181 °C, respectively. Samples NS2 and NS4 are the only ones that contain lactose monohydrate. Lactose monohydrate has two weight losses related to water loss. As research shows, the first mass loss occurs at a temperature between 130 and 170 °C (with a maximum at 150 °C) and is associated with the loss of surface water; the second at a temperature of about 220 °C is associated with the loss of water of crystallization [26,42,48,49]. The recorded peak likely comes from lactose monohydrate. Although the peak values recorded for the NS2 and NS4 samples are slightly higher than for lactose monohydrate, Altamimi M. J. et al. show that lactose is added pharmaceutical preparations by many manufacturers and is often in the form of anomers (α, and β) [50]. The anomers differ in their physical properties [50,51]. The authors showed differences in the anomer composition of 19 commercially available lactose monohydrates on the market and showed the need to monitor lactose composition in terms of its anomers in the pharmaceutical industry [50]. The last peak from the excipient, with a maximum weight loss between 301 °C (NS4) and 312.2 °C (NS2), was recorded for all tested pharmaceutical preparations. This peak is likely from microcrystalline cellulose. Due to the different types of cellulose, it is characterized by a different temperature of maximum mass loss. However, this peak cannot come from other common excipients in the study drugs, such as magnesium stearate, povidone, and talc, because the maximum weight loss for these excipients occurs at much higher temperatures [47]. For magnesium stearate, it is 370–395 °C [26,27,41,42,43], for povidone 434–485 °C [27,41,52], and for talc above 800 °C [41,53].

Figure 7 and Table 7 present c-DTA curves and their parameters registered for the naproxen sodium salt standard and pharmaceutical products containing NSS as API in the formulation. The naproxen salt has three events on the c-DTA curve compared to naproxen acid. The first and second peaks are endothermic and have a maximum of 72.7 °C and 257.4 °C, respectively. The first endothermic peak is associated with the release of water and correlates well with the TG/DTG/D2TG curves (Figure 6) [23,33,45]. The second endothermic peak is associated with the melting peak of the sample [33,41]. The melting point of the naproxen salt is shifted by 98.7 °C towards a higher temperature compared to the naproxen acid. This clearly indicates a higher thermal resistance of naproxen sodium. The last peak, with a maximum of 412.8 °C, is exothermic and related to the thermal degradation of naproxen sodium salt [33,40]. This peak corresponds with thermogravimetric curves (Figure 6).

The melting point and the exothermic peak associated with the decomposition of naproxen are similar for all tested pharmaceutical preparations containing naproxen salt. For the tested compounds, the maximum peak of the melting point is in the range of 239.4 °C (NS4) to 254.9 °C (NS3). The maximum exothermic peak for the tested drugs occurred between 408.4 °C (NS1) and 418.7 °C (NS4).

The endothermic peak was recorded on the c-DTA curves for drugs NS1–NS5. The maximum peak for pharmaceutical products was between 41.5 °C (NS4) and 91.7 °C (NS2). These peaks were related to the release of water from the API and had the most extensive spread regarding the maximum peak temperature. In addition, a second endothermic peak was recorded for the sample NS2, with a maximum of 58 °C. This peak is related to the release of water from microcrystalline cellulose and is more pronounced than other drugs.

### 3.2. FTIR and DFT Analysis

Figure 8 shows the FTIR analysis of (a) naproxen sodium salt and (b) naproxen acid (4000–2600 cm^−1^). Regarding naproxen sodium (Figure 8a), the FTIR spectra show several differences between the commercial products. The main differences are related to the range of ~3600 to ~3100 cm^−1^. There is a similarity between the theoretical (density functional theory, DFT) spectra and experimentally obtained ones from the commercially available standard due to the absence (Figure 8a) and presence (Figure 8b, in the case of the DFT spectrum, the peak is shifted towards larger wavenumbers ~3700 cm^−1^) of the hydroxyl functional group (in carboxylate/carboxylic acid part of the compound). The band at ~3455 cm^−1^ increases the intensity when the water content increases due to the O−H moiety vibration. Similar outcomes were reported previously by Jamrógiewicz and colleagues [54]. At the same time, the band is marginally visible in the naproxen standard. It should be noted that the presence of a band at ~3315 cm^−1^ in the sample named Nalgesin (NS1) is characteristic for the monohydrate and dihydrate form [54]. According to the literature [54], intensity decreases with higher water content. In the same way, the bands at ~3016 cm^−1^ and ~2976 cm^−1^, which show C-H bonds, decrease in intensity with higher content of water.

On the other hand, Figure 8b shows a band at ~3215 cm^−1^, which corresponds to O-H from the carboxylic group and water [55], while at ~3002 cm^−1^, the band of C-H (aromatic part) appears [55]. Both bands were present in all analyzed samples.

Figure 9 presents FTIR spectra in the range of 2000–500 cm^−1^ of naproxen sodium salt (Figure 9a) and naproxen acid (Figure 9b). Regarding Figure 9a, based on the literature, the most important bands for naproxen identification were selected [55]. At a wavenumber of 1600 cm^−1^ and 1450 cm^−1^, the bands corresponding to C=C of the polycyclic aromatic structure and C-H were identified. While, at 1390 cm^−1^, the signal was ascribed to O-H. Continuing, the bands at 1265 cm^−1^ and 1209 cm^−1^ were ascribed to C-O, as reported previously [55,56].

On the other hand, the spectra of acid naproxen (Figure 9a) show several bands; the signal ascribed to C=O from carboxylic acids is present at 1720 cm^−1^. As for Figure 9a, at 1600 cm^−1^, the signal refers to C=C (polycyclic aromatic). Continuing with the spectra, at 1450 cm^−1^, the signal was ascribed to C-H (aromatic structure). Similarly to Figure 8, the most significant difference between the naproxen sodium salt and acid is in the carboxylic/carboxylate group region, where the vibration of the C=O group is visible at larger wavenumbers of 1800–1700 cm^−1^ in naproxen acid, both in the theoretical and experimental spectra (Figure 9b).

### 3.3. UV-Vis Spectrophotometry and HPLC Analysis

UV-Vis and HPLC methods were used to identify API in the solutions containing naproxen in acid and sodium salt form. Both techniques are proposed by the pharmacopoeia to identify API in pharmaceutical samples [9,10,11].

For the tested standards of naproxen (NAS, NSS) and pharmaceutical preparations containing naproxen acid (NA1–NA4) and naproxen sodium salt (NS1–NS5), UV absorbance spectra registration was performed under the same conditions. The recording was carried out for samples dissolved in methanol in the wavelength range of 245 to 345 nm (Figure 10). The conducted research shows that naproxen both in the form of acid and sodium salt shows four absorbance maxima occurring at the same wavelengths. The wavelength for absorbance maximum 1–4 is 262, 271, 316, and 331 nm, respectively [9]. For drugs containing naproxen sodium salt as an API, a shift of the absorbance maximum for a peak at 316 nm and peak at 331 nm by 1 nm towards a higher wavelength can be observed. However, this is too little visible information to determine the type of API in the formulation.

This indicates a major limitation of this method for identifying APIs in an unknown formulation. The fact of the high similarity of the spectra is due to the small influence of the substituents and fragments of the saturated molecule on the shape of the absorption curve of the basic compound [57].

HPLC analysis is shown in Figure 11 (naproxen sodium) and Figure 12 (naproxen acid). A separate calibration was made for each, based on pure APIs. Regarding Figure 11, elution time slightly changes between the samples (4.23–4.37 min), while naproxen acid shows almost identical elution time (4.2 min). Due to the overlap of retention times for both types of naproxen, using this method as a standard method to distinguish NA sodium and NA acid is challenging since the concentration of each species will depend on the pH value of the mobile phase.

UV-Vis spectrophotometry and the HPLC method are effective only when we know which API is in the drug. Then, we can identify the API in the pharmaceutical preparation using a specific pattern.

### 3.4. Colorimetric Analysis

In addition, in the experiment, colorimetric analyses were performed in the CIE L*a*b* system to assess the color of the tested samples. The studies were performed in order to show significant differences in the color of the naproxen standards and tested drugs. Figure 13 shows the color of naproxen acid standard and preparations containing NA as an API in the 3D CIE L*a*b* space. In turn, Figure 14 shows the same relationships as Figure 15 only for naproxen sodium salt standard and pharmaceutical preparations that contain it. The results obtained for naproxen acid and sodium salt indicate the location of the analyzed parameters are in very close proximity. This indicates minimal differences in the color of the analyzed drugs and their similarity to the color of the used naproxen standards. This may additionally confirm the use of the correct API in the formulation. However, using colorimetry, we cannot distinguish the form of naproxen used in the pharmaceutical preparations, because both naproxen standards (NAS, NSS) show too-similar values of the L*, a* and b* color space parameters.

### 3.5. Evaluation of the Specificity of the TGA and c-DTA Method

Nabumetone (NAB) was used to assess the specificity of the applied thermal methods. Nabumetone was used for the study due to the fact that it is a close structural analogue of naproxen [2,5]. Nabumetone is a prodrug with a ketone structure belonging to NSAIDs (Figure 15a) [5]. It is rapidly absorbed after oral administration. As a result of the first pass, it is converted into an active metabolite ((6-methoxy-2-naphthyl)acetic acid) in the liver (Figure 15b) [2,5]. (6-methoxy-2-naphthyl)acetic acid is a lower homolog of naproxen, showing strong inhibition of cyclooxygenase, mainly isoform 2 (COX-2) [2,7].

The TG curve of the nabumetone shows three steps of thermal decomposition. The major mass loss in the first stage comprises 76.07% (Figure 16). The second and third mass losses comprise 6.20% and 9.11%, respectively. The onset temperature of decomposition nabumetone is 230.1 °C. This shows that despite the similar structure to naproxen, the onset temperature of the decomposition of nabumetone is lower by 24.9 °C and 145.9 °C compared to NAS and NSS, respectively.

The DTG and D2TG curves of nabumetone presented three peaks corresponding with the TG curve (Figure 17). The DTG major first stage of mass loss occurred in the temperature range of 153 °C to 274 °C, with a maximum peak of 257.4 °C (−22.38%/min). This stage is related to a significant decomposition of nabumetone. Despite the close structural similarity of nabumetone to naproxen, using thermogravimetry, we record a shift of the main stage of decomposition towards lower temperatures (257.4 °C). This allows for an unambiguous distinction between these compounds. The second stage on the DTG curve occurred in the temperature range of 274 °C to 319 °C with a maximum peak of 298 °C and mass change of −2.07%/min. The last stage for nabumetone begins at 319 °C, with maximum mass loss at 334 °C (−1.49%/min.). All registered peaks on DTG/D2TG curves for nabumetone were associated with thermal decomposition.

Similarly to the recorded TG, DTG, and D2TG curves, the c-DTA curve also showed significant differences for nabumetone compared to the naproxen standards (NAS, NAS). Two peaks were recorded for nabumetone on the c-DTA curve (Figure 18). The first peak is endothermic, with a maximum of 85.3 °C. This peak is associated with the melting of nabumetone and occurs 73.5 °C and 172.3 °C earlier than the NAS and NSS melting peaks, respectively [33,41]. The second peak is exothermic and has a maximum of 271.3 °C. This peak is related to the decomposition of nabumetone and starts early compared to the decomposition peaks of naproxen standards [33].

In addition, UV-Vis spectrophotometry was used to compare the results obtained by thermal analyses (TGA, c-DTA). UV absorbance spectra in the wavelength range of 245 to 345 nm were determined for the tested nabumetone and naproxen standards (NAS, NSS). The obtained spectra did not allow us to distinguish nabumetone due to the occurrence of absorbance maxima at the same points as the NA and NS standards (Figure 19).

The research confirmed the specificity and usefulness of TGA and c-DTA methods for distinguishing close analogues of naproxen. This indicates thermal methods’ applicability for analyzing, identifying, and differentiating API in pharmaceutical formulations.

## 4. Conclusions

The research has shown that thermogravimetry (TGA) assisted by calculated differential thermal analysis (c-DTA) helps distinguish between two forms of naproxen-acid and sodium salts. This allows for the initial identification of the type of naproxen form in commercial pharmaceutical preparations.

TGA in combination with c-DTA makes it possible to distinguish naproxen in the form of acid or sodium salt by registering specific thermal events for each form, which is not possible with some pharmacopoeial methods such as UV-Vis or HPLC. For some pharmaceutical preparations, it is also possible to record thermal events from excipients.

In addition, using a close structural analog of naproxen (nabumetone), high specificity for the evaluated compounds of the applied thermal methods was demonstrated.

Research has shown that TGA supported by c-DTA is as effective in determining the type of naproxen form in various pharmaceutical preparations as the commonly used FTIR technique.

However, it should be remembered that despite many advantages, such as speed and simplicity of measurement, small sample volume, easy preparation, repeatability, and low cost of measurement, the applied thermal methods have some limitations. For example, in order to obtain high repeatability, the measurements must be made under the same conditions, using the same measuring crucible, sample weight and, often, the same apparatus.

## Figures and Tables

**Figure 1 pharmaceutics-15-01689-f001:**
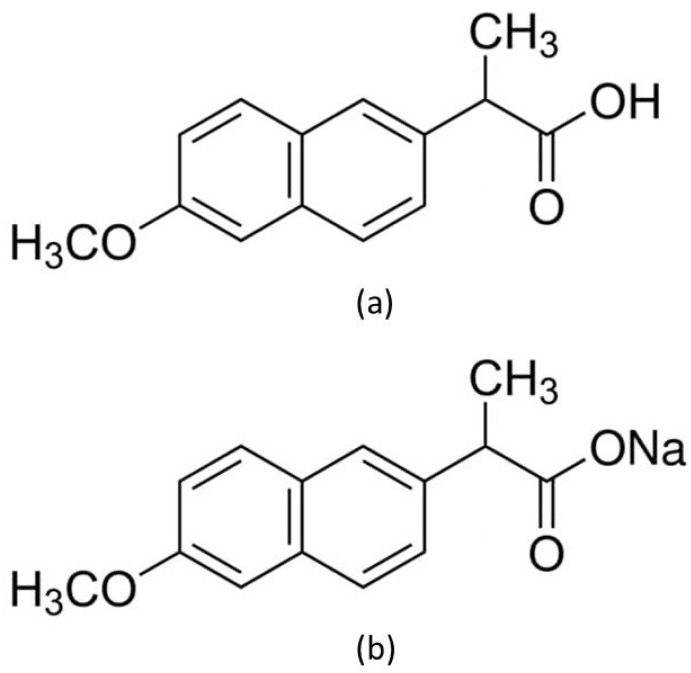
Chemical structure of (**a**) naproxen acid, and (**b**) naproxen sodium salt [5].

**Figure 2 pharmaceutics-15-01689-f002:**
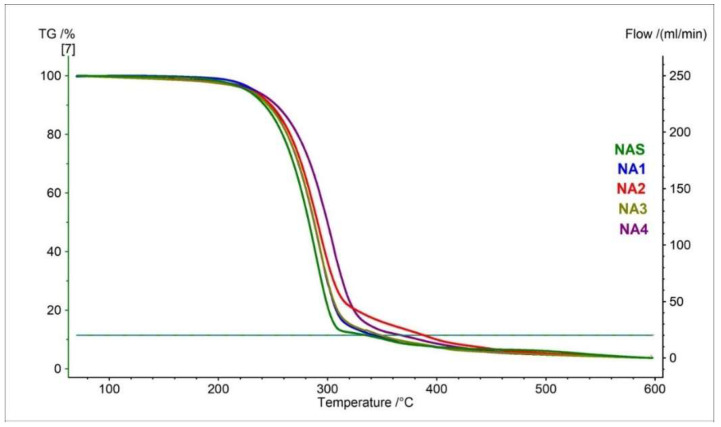
TG curves of naproxen acid standard (NAS) and pharmaceutical preparations containing naproxen acid (NA1–NA4).

**Figure 3 pharmaceutics-15-01689-f003:**
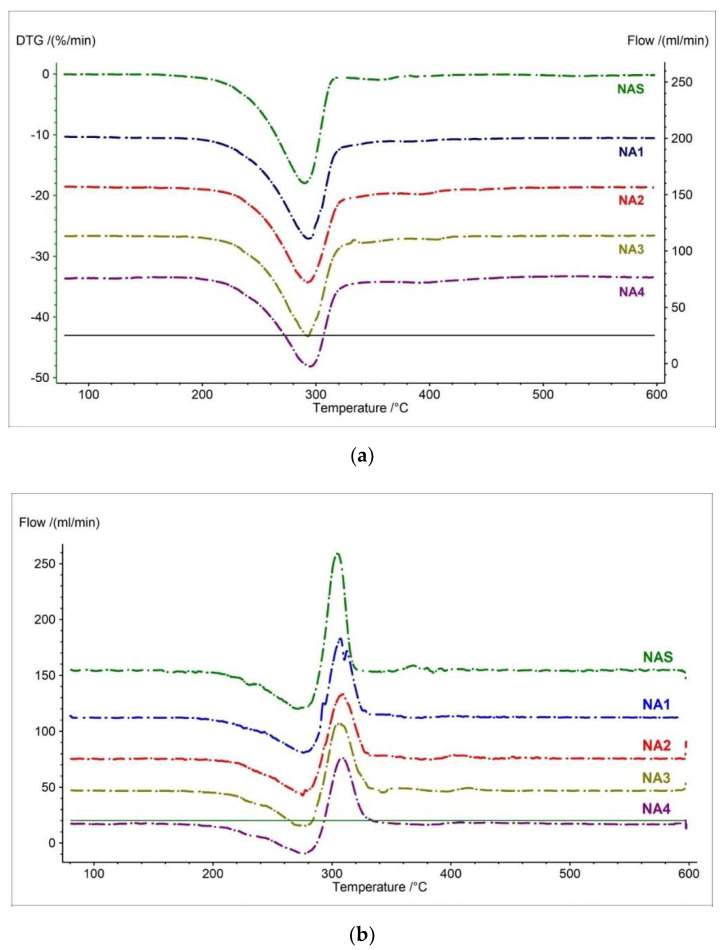
(**a**) DTG and (**b**) D2TG curves of naproxen acid standard (NAS) and pharmaceutical preparations containing naproxen acid (NA1–NA4).

**Figure 4 pharmaceutics-15-01689-f004:**
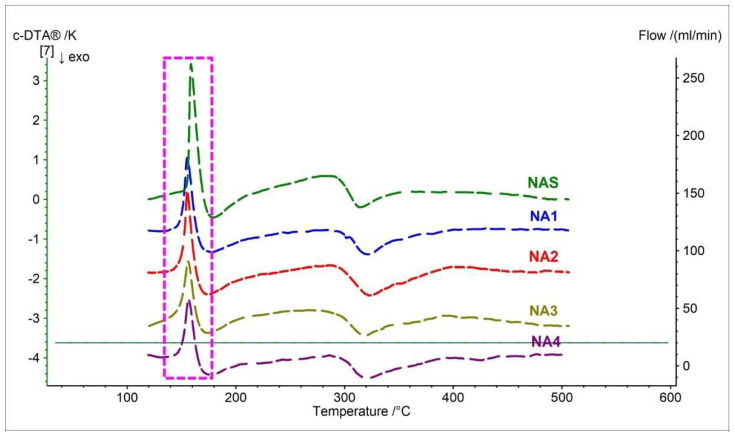
c-DTA curves of naproxen acid standard (NAS) and pharmaceutical preparations containing naproxen acid (NA1–NA4). The dashed frame indicates the melting point.

**Figure 5 pharmaceutics-15-01689-f005:**
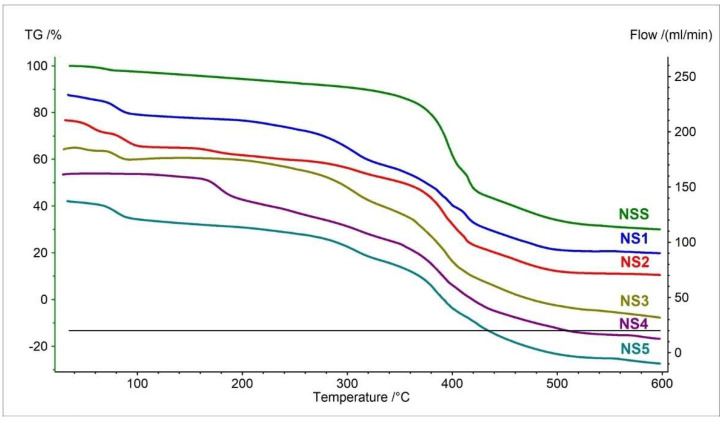
TG curves of naproxen sodium standard (NSS) and pharmaceutical preparations containing naproxen sodium salt (NS1–NS5).

**Figure 6 pharmaceutics-15-01689-f006:**
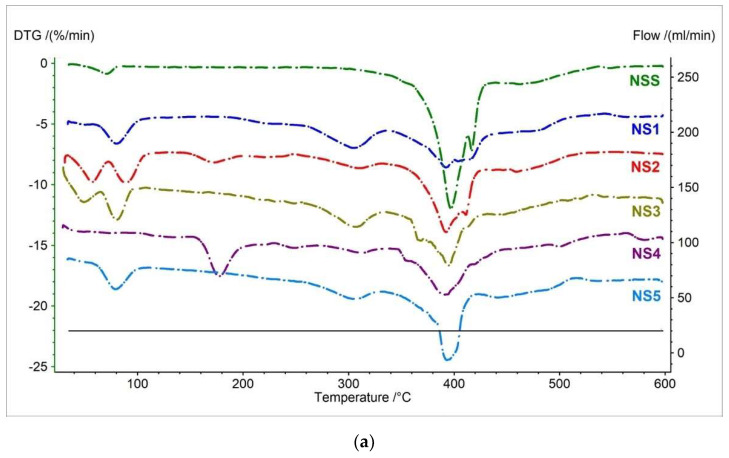
(**a**) DTG and (**b**) D2TG curves of naproxen sodium standard (NSS), and pharmaceutical preparations containing naproxen sodium salt (NS1–NS5).

**Figure 7 pharmaceutics-15-01689-f007:**
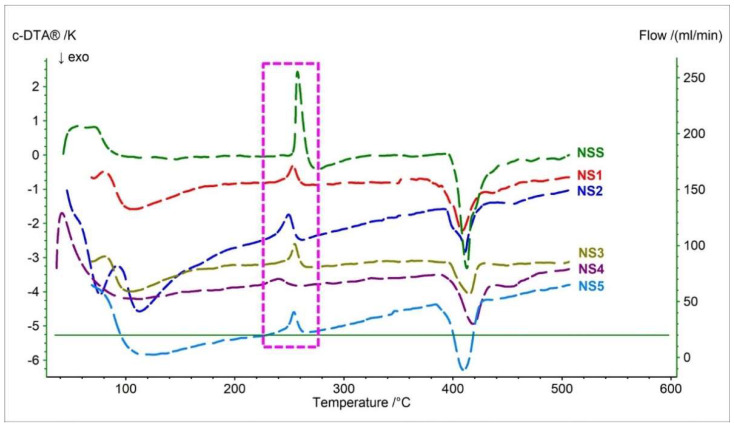
c-DTA curves of naproxen sodium standard (NSS) and pharmaceutical preparations containing naproxen sodium salt (NS1–NS5). The dashed frame indicates the melting point.

**Figure 8 pharmaceutics-15-01689-f008:**
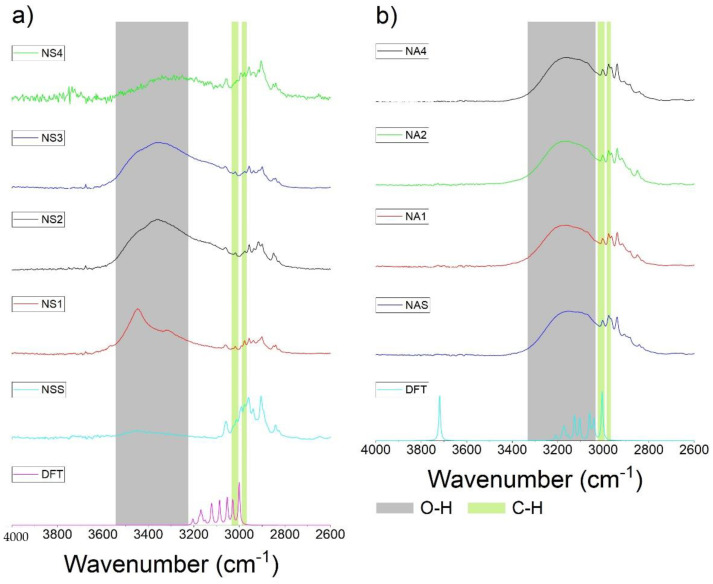
FTIR spectra of naproxen in (**a**) sodium salt form and (**b**) acid form; wavenumber: 4000–2600 cm^−1^. Spectra obtained by density functional theory (DFT) were compared to the experimental data.

**Figure 9 pharmaceutics-15-01689-f009:**
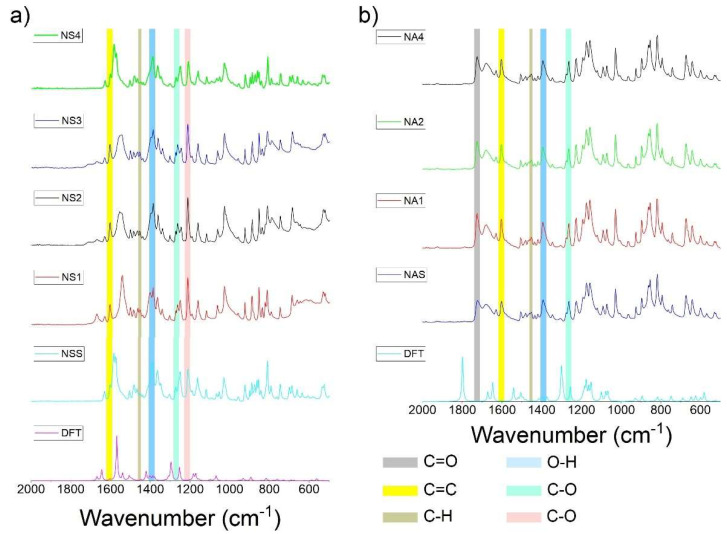
FTIR spectra of naproxen in (**a**) sodium salt form and (**b**) acid form; wavenumber: 2000–500 cm^−1^. Spectra obtained by density functional theory (DFT) were compared to the experimental data.

**Figure 10 pharmaceutics-15-01689-f010:**
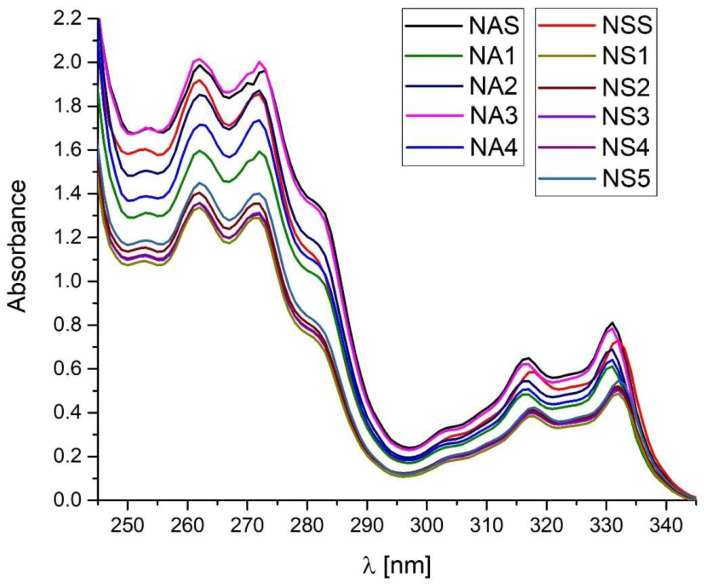
UV absorbance spectra of naproxen standards (NAS, NSS) and tested pharmaceutical preparations containing naproxen acid (NA1–NA4) and naproxen sodium salt (NS1–NS5). Spectra recorded in the range of 245 nm to 345 nm at room temperature.

**Figure 11 pharmaceutics-15-01689-f011:**
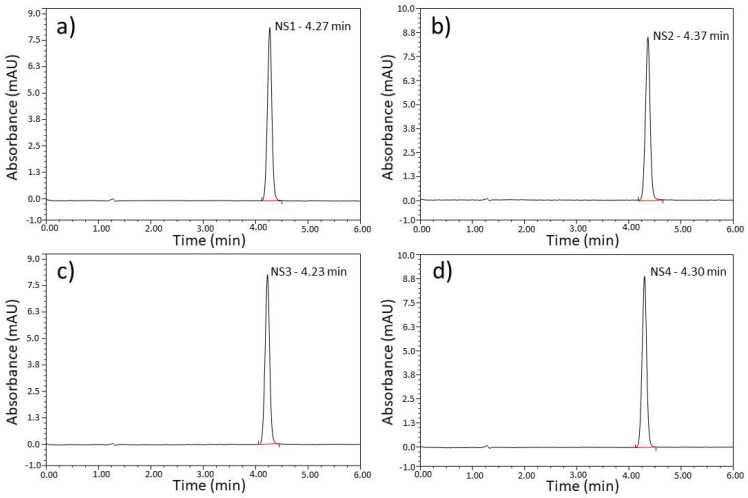
HPLC analysis of pharmaceutical preparations containing naproxen sodium (**a**) NS1, (**b**) NS2, (**c**) NS3, and (**d**) NS4.

**Figure 12 pharmaceutics-15-01689-f012:**
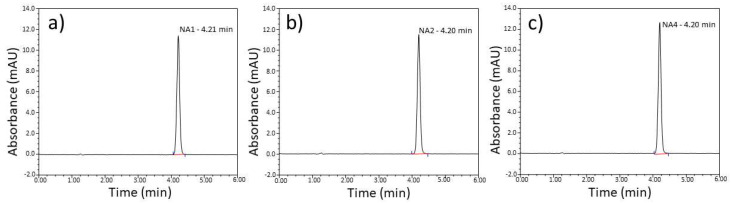
HPLC analysis of pharmaceutical preparations containing naproxen acid (**a**) NA1, (**b**) NA2, and (**c**) NA4. X-axis = elution time, y-axis = intensity.

**Figure 13 pharmaceutics-15-01689-f013:**
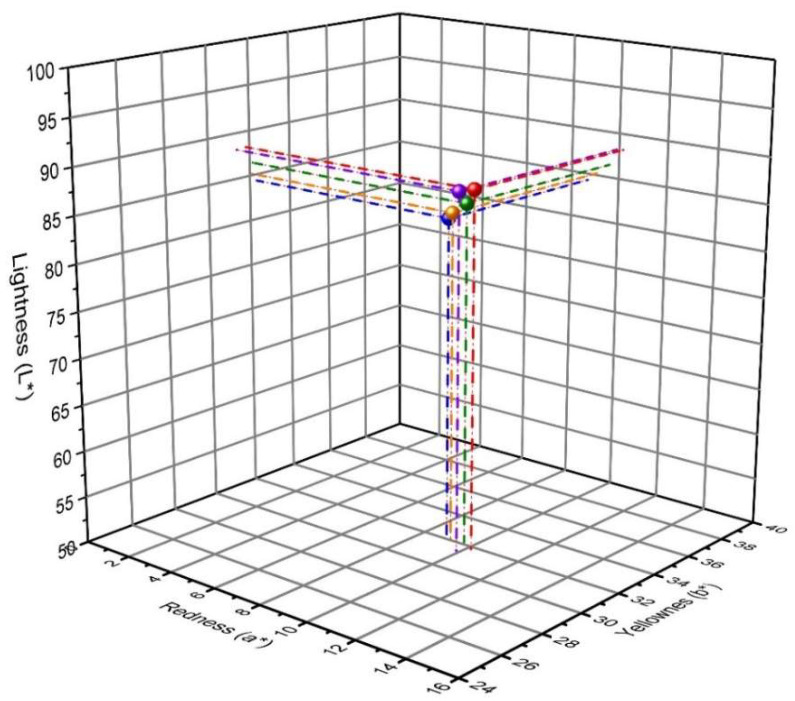
The colorimetric analysis in the 3D CIE L*a*b* space of naproxen acid standard (green) and pharmaceutical preparations containing naproxen acid: NA1 (red), NA2 (blue), NA3 (yellow), and NA4 (purple). The results are presented as mean ± SD (*n* = 8).

**Figure 14 pharmaceutics-15-01689-f014:**
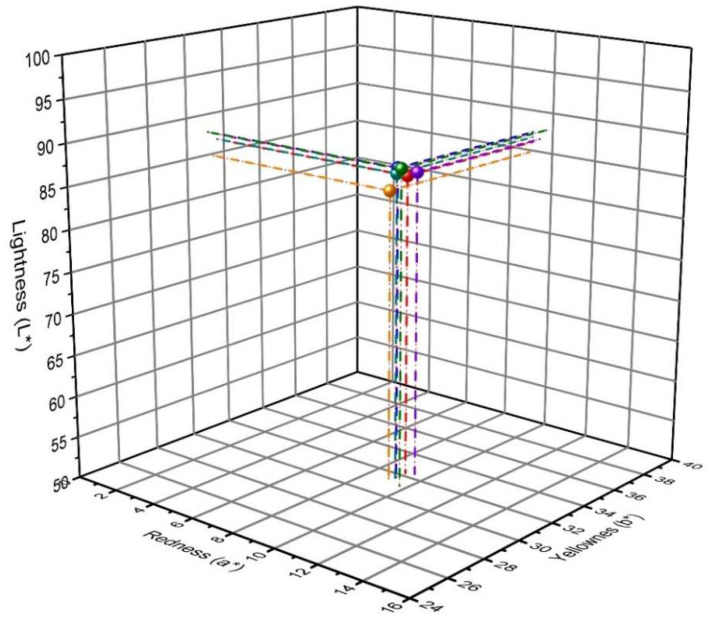
The colorimetric analysis in the 3D CIE L*a*b* space of naproxen sodium standard (green) and pharmaceutical preparations containing naproxen sodium: NS1 (red), NS2 (navy blue), NS3 (yellow), NS4 (purple), and NS5 (blue). The results are presented as mean ± SD (*n* = 8).

**Figure 15 pharmaceutics-15-01689-f015:**
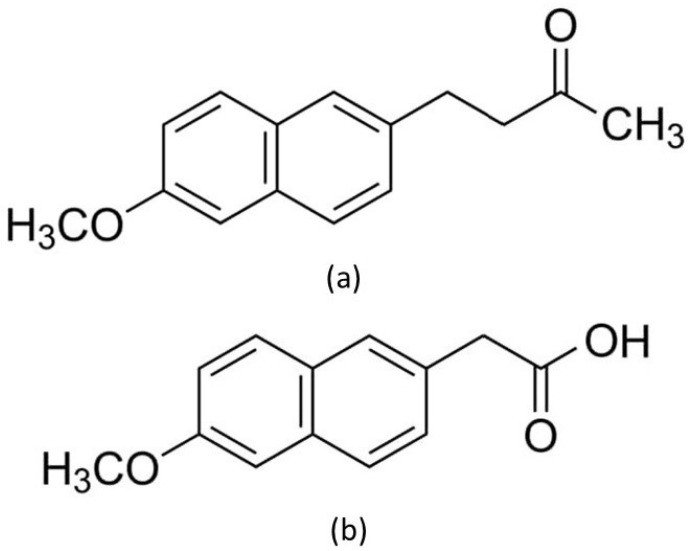
Chemical structure of (**a**) nabumetone and (**b**) active metabolite of nabumetone (6-methoxy-2-naphthyl)acetic acid [5,7].

**Figure 16 pharmaceutics-15-01689-f016:**
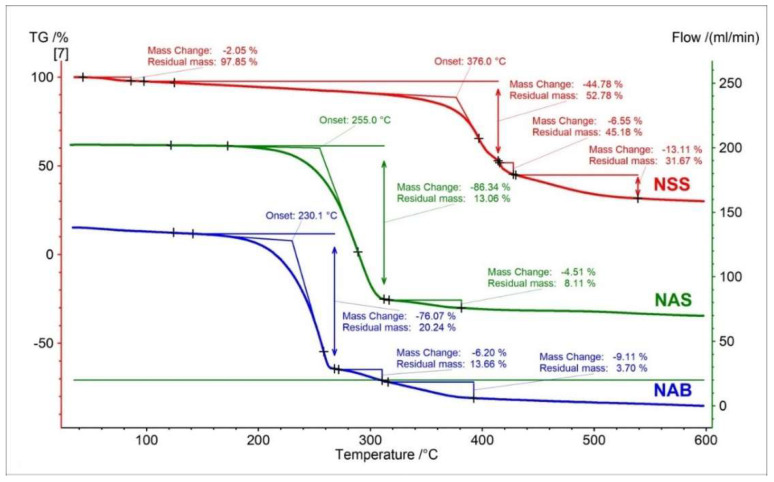
TG curves of naproxen sodium standard (NSS), naproxen acid standard (NSA), and nabumetone (NAB).

**Figure 17 pharmaceutics-15-01689-f017:**
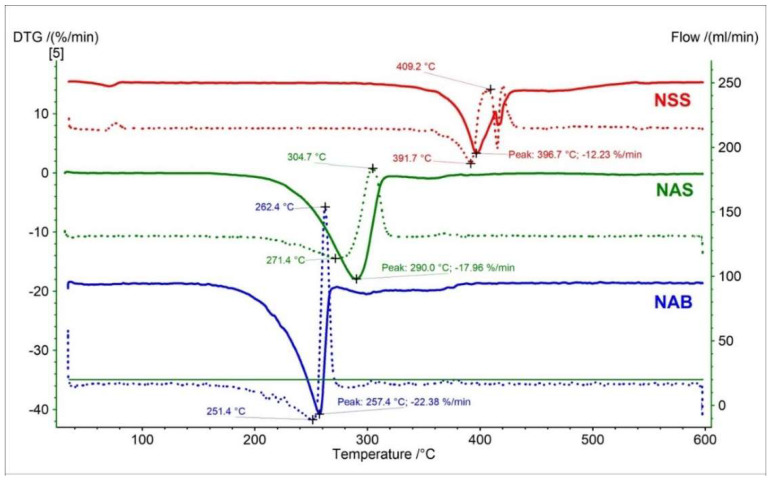
DTG (solid line) and D2TG (dotted line) curves of naproxen sodium standard (NSS), naproxen acid standard (NSA), and nabumetone (NAB).

**Figure 18 pharmaceutics-15-01689-f018:**
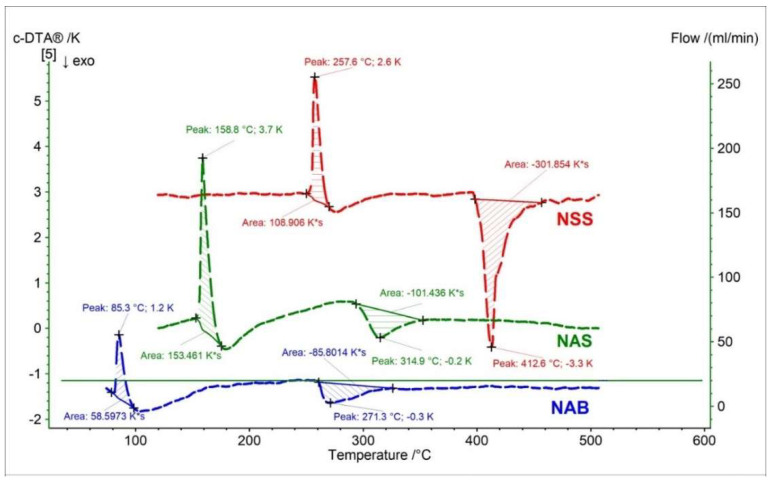
c-DTA curves of naproxen sodium standard (NSS), naproxen acid standard (NSA), and nabumetone (NAB).

**Figure 19 pharmaceutics-15-01689-f019:**
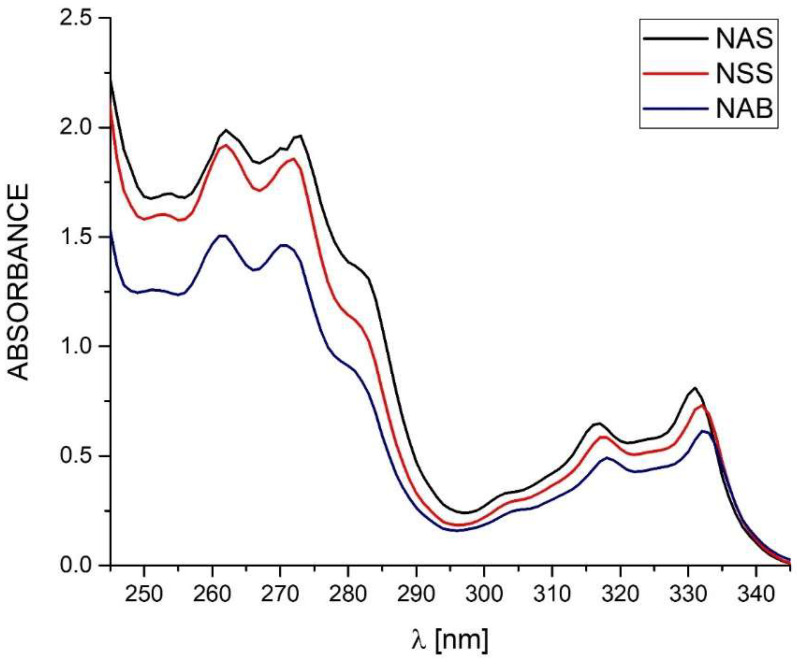
UV absorbance spectra of naproxen acid standard (NAS), naproxen sodium salt standard (NSS), and nabumetone (NAB). Spectra recorded in the range of 245 nm to 345 nm at room temperature.

**Table 1 pharmaceutics-15-01689-t001:** Tested commercially available drugs containing different forms of naproxen. POM: Prescription Only Medicine, OTC: Over-The-Counter.

Samples	Pharmaceutical Preparation, Drug Form, Producer, Drug Availability	API, Dose, Weight of Tablet	Excipients (Tablet Core)	Macroscopic Appearance of the Tablet
NA1	Naproxen 250 Hasco, tablet, Hasco-Lek (Poland), POM	Naproxen, 250 mg, 272.9 mg	Povidone K90 Croscarmellose sodium Magnesium stearate	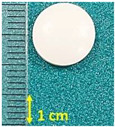
NA2	Naproxen 500 Hasco, tablet, Hasco-Lek (Poland), POM	Naproxen, 500 mg, 543.8 mg	Povidone K90 Croscarmellose sodium Magnesium stearate	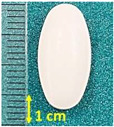
NA3	Apo-Napro, tablet, Apotex Europea B.V. (Netherlands), POM	Naproxen, 250 mg, 275.3 mg	Methylcellulose Croscarmellose sodium Magnesium stearate Colloidal anhydrous silica	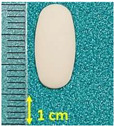
NA4	Anapran EC, gastro-resistant tablets, Adamed Pharma S.A. (Poland), POM	Naproxen, 250 mg, 304.0 mg	Povidone K90 Croscarmellose sodium Magnesium stearate	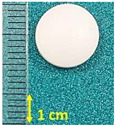
NS1	Nalgesin, coated tablets, Krka (Slovenia), POM	Naproxen Sodium, 275 mg, 434.4 mg	Povidone K30 Microcrystalline cellulose Talc Magnesium stearate	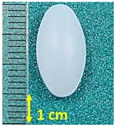
NS2	Anapran, coated tablets, Adamed Pharma S.A. (Poland), POM	Naproxen Sodium, 275 mg, 402.8 mg	Povidone Microcrystalline cellulose Talc Magnesium stearate Hydroxypropyl cellulose Lactose monohydrate Stearic acid Hypromellose Macrogol 8000	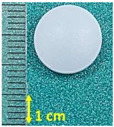
NS3	Alleve, coated tablets, Bayer Bitterfeld GmbH (Germany), OTC	Naproxen Sodium, 220 mg, 322.6 mg	Povidone K30 Microcrystalline cellulose Talc Magnesium stearate	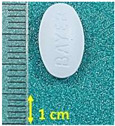
NS4	Naxii, coated tablets, US Pharmacia (Poland), OTC	Naproxen Sodium, 220 mg, 324.5 mg	Povidone Microcrystalline cellulose Talc Magnesium stearate Lactosum monohydrate Colloidal anhydrous silica	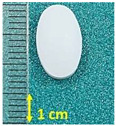
NS5	Nalgesin Mini, coated tablets, Krka (Slovenia), OTC	Naproxen Sodium, 220 mg, 319.8 mg	Povidone K30 Microcrystalline cellulose Talc Magnesium stearate	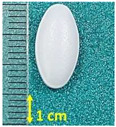

**Table 2 pharmaceutics-15-01689-t002:** Characteristic parameters of thermogravimetry curves (TGA) of the naproxen acid standard (NAS) and pharmaceutical preparations containing naproxen acid (NA1–NA4).

Tested Samples	TG Parameters				
	Onset (°C)	Mid (°C)	Inflection (°C)	End (°C)	Weight Loss (%)
NAS	255.0	281.0	288.8	307.6	−89.33
NA1	258.1	285.7	293.7	313.4	−85.44
NA2	259.3	287.8	293.0	316.8	−84.86
NA3	258.7	286.2	293.3	313.7	−83.73
NA4	257.5	286.4	294.7	315.9	−80.93

**Table 3 pharmaceutics-15-01689-t003:** Characteristic parameters of DTG and D2TG curves of the naproxen acid standard (NAS) and pharmaceutical preparations containing naproxen acid (NA1–NA4). Decomposition stages related to pure API are in bold.

Tested Samples	Type of Curve	Parameters	Stage	
			I	II
NAS	DTG	Peak (°C)	**290.0**	**356.0**
		Mass change (%/min)	**−17.96**	**−0.93**
	D2TG	Peak min. (°C)	**271.4**	**341.0**
		Peak max. (°C)	**304.7**	**368.0**
NA1	DTG	Peak (°C)	**293.2**	**383.0**
		Mass change (%/min)	**−16.67**	**−0.68**
	D2TG	Peak min. (°C)	**276.2**	**374.0**
		Peak max. (°C)	**308.7**	**401.0**
NA2	DTG	Peak (°C)	**292.0**	**393.0**
		Mass change (%/min)	**−15.82**	**−1.31**
	D2TG	Peak min. (°C)	**279.2**	**385.0**
		Peak max. (°C)	**309.2**	**406.0**
NA3	DTG	Peak (°C)	**292.3**	**403.0**
		Mass change (%/min)	**−16.46**	**−0.73**
	D2TG	Peak min. (°C)	**275.2**	**401.0**
		Peak max. (°C)	**306.8**	**412.0**
NA4	DTG	Peak (°C)	**294.6**	**390.7**
		Mass change (%/min)	**−14.96**	**−1.18**
	D2TG	Peak min. (°C)	**280.7**	**382.0**
		Peak max. (°C)	**307.8**	**406.0**

**Table 4 pharmaceutics-15-01689-t004:** Characteristic parameters c-DTA of naproxen acid standard (NAS) and pharmaceutical preparations containing naproxen acid (NA1–NA4). Information on the melting point is in bold. Endo = endothermic reaction, Exo = exothermic reaction.

Tested Samples	Parameter	Peak	
		I	II
NAS	Onset (°C)	**155.8**	297.0
	Peak (°C)	**158.7**	314.5
	Type of reaction	**Endo**	Exo
	Area (K*s)	**154.9**	103.5
NA1	Onset (°C)	**150.3**	298.4
	Peak (°C)	**155.3**	321.8
	Type of reaction	**Endo**	Exo
	Area (K*s)	**105.7**	114.0
NA2	Onset (°C)	**150.7**	283.6
	Peak (°C)	**155.3**	323.2
	Type of reaction	**Endo**	Exo
	Area (K*s)	**119.0**	224.4
NA3	Onset (°C)	**149.3**	298.4
	Peak (°C)	**156.1**	318.7
	Type of reaction	**Endo**	Exo
	Area (K*s)	**102.6**	143.5
NA4	Onset (°C)	**150.7**	301.2
	Peak (°C)	**156.7**	320.0
	Type of reaction	**Endo**	Exo
	Area (K*s)	**108.2**	138.8

**Table 5 pharmaceutics-15-01689-t005:** Characteristic parameters of thermogravimetry curves (TGA) of naproxen sodium standard (NSS) and pharmaceutical preparations containing naproxen sodium salt (NS1–NS5).

Tested Samples	TG Parameters				
	Onset (°C)	Mid (°C)	Inflection (°C)	End (°C)	Weight Loss (%)
NSS	375.6	359.9	395.5	424.2	−32.20
NS1	342.6 * 373.9 **	389.4	411.6	426.5	−27.91
NS2	354.2 * 375.3 **	375.9	392.8	409.4	−31.63
NS3	312.6 * 373.9 **	350.6	390.1	423.5	−29.29
NS4	340.6 * 369.5 **	375.3	382.5	420.3	−32.49
NS5	330.6 * 375.6 **	359.9	395.5	424.2	−32.20

* Decomposition onset of formulation. ** Decomposition onset of API in formulation.

**Table 6 pharmaceutics-15-01689-t006:** Characteristic parameters of DTG and D2TG curves of naproxen sodium standard (NSS) and pharmaceutical preparations containing naproxen sodium salt (NS1–NS5). Decomposition stages related to pure API are in bold.

Tested Samples	Type of Curve	Parameters	Stage						
			I	II	III	IV	V	VI	VII
NSS	DTG	Peak (°C)	**70.6**	**396.7**	**416.0**	**462.6**	-	-	-
		Mass change (%/min)	**−0.89**	**−12.23**	**−7.47**	**−1.73**	-	-	-
	D2TG	Peak min. (°C)	**65.0**	**391.7**	**415.4**	**459.4**	-	-	-
		Peak max. (°C)	**76.9**	**409.2**	**420.7**	**494.9**	-	-	-
NS1	DTG	Peak (°C)	49.0	**80.0**	308.0	**397.0**	**413.0**	**460.0**	-
		Mass change (%/min)	−0.90	**−2.42**	−2.93	**−4.98**	**−4.74**	**−1.45**	-
	D2TG	Peak min. (°C)	40.0	**72.4**	276.3	**384.9**	**409.9**	**450.5**	-
		Peak max. (°C)	55.0	**88.8**	316.2	400.9	**418.2**	**484.2**	-
NS2	DTG	Peak (°C)	57.2	**88.4**	172.3	312.2	**392.5**	**411.3**	**458.9**
		Mass change (%/min)	−2.55	**−2.61**	−0.94	−1.40	**−6.66**	**−5.24**	**−1.70**
	D2TG	Peak min. (°C)	50.0	**79.1**	159.1	303.0	**386.5**	**408.0**	**456.0**
		Peak max. (°C)	64.7	**99.2**	180.0	320.1	**396.8**	**414.5**	**464.0**
NS3	DTG	Peak (°C)	48.8	**81.2**	303.5	**394.4**	**417.0**	**461.0**	-
		Mass change (%/min)	−0.78	**−2.90**	−2.01	**−6.85**	**−2.96**	**−1.62**	-
	D2TG	Peak min. (°C)	39.0	**72.0**	288.8	**381.0**	**415.0**	**457.0**	-
		Peak max. (°C)	53.0	**87.0**	317.0	**399.0**	**424.0**	**497.0**	-
NS4	DTG	Peak (°C)	**68.1**	181.0	301.0	**388.6**	**423.5**	**466.0**	-
		Mass change (%/min)	**−0.32**	−4.05	−1.16	**−5.50**	**−2.30**	**−1.32**	-
	D2TG	Peak min. (°C)	**50.0**	170.0	279.0	**384.0**	**419.0**	**454.1**	-
		Peak max. (°C)	**77.0**	184.0	305.2	**399.1**	**427.2**	**470.0**	-
NS5	DTG	Peak (°C)	52.0	**81.0**	308.0	**393.0**	**425.0**	**457.0**	-
		Mass change (%/min)	−0.54	**−2.83**	−2.19	**−4.98**	**−2.99**	**−1.73**	-
	D2TG	Peak min. (°C)	49.0	**70.0**	286.0	**379.0**	**422.0**	**455.0**	-
		Peak max. (°C)	56.0	**89.0**	317.0	**400.0**	**430.0**	**488.0**	-

**Table 7 pharmaceutics-15-01689-t007:** Characteristic parameters c-DTA of naproxen sodium standard (NSS) and pharmaceutical preparations containing naproxen sodium salt (NS1–NS5). Information on the melting point is in bold. Endo = endothermic reaction, Exo = exothermic reaction.

Tested Samples	Parameter	Peak			
		I	II	III	IV
NSS	Onset (°C)	64.2	**253.6**	401.4	-
	Peak (°C)	72.7	**257.4**	412.8	-
	Type of reaction	Endo	**Endo**	Exo	-
	Area (K*s)	11.7	**106.2**	302.1	-
NS1	Onset (°C)	71.4	**247.9**	397.0	-
	Peak (°C)	80.2	**253.4**	408.4	-
	Type of reaction	Endo	**Endo**	Exo	-
	Area (K*s)	42.8	**28.5**	217.3	-
NS2	Onset (°C)	-	76.7	**240.3**	388.6
	Peak (°C)	58.0	91.7	**249.3**	411.3
	Type of reaction	Endo	Endo	**Endo**	Exo
	Area (K*s)	30.3	108.0	**44.0**	110.9
NS3	Onset (°C)	72.3	**250.5**	398.0	-
	Peak (°C)	80.5	**254.9**	414.7	-
	Type of reaction	Endo	**Endo**	Exo	-
	Area (K*s)	47.4	**23.1**	82.9	-
NS4	Onset (°C)	38.5	**227.4**	401.6	-
	Peak (°C)	41.5	**239.4**	418.7	-
	Type of reaction	Endo	**Endo**	Exo	-
	Area (K*s)	34.9	**15.1**	250.3	-
NS5	Onset (°C)	-	**249.5**	396.8	-
	Peak (°C)	78.0	**254.1**	409.7	-
	Type of reaction	Endo	**Endo**	Exo	-
	Area (K*s)	24.4	**21.5**	223.6	-

## Data Availability

The data that support the findings of this study are available from the corresponding author upon reasonable request.

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
