# Peer review of "Application of TGA/c-DTA for Distinguishing between Two Forms of Naproxen in Pharmaceutical Preparations"

_pharmaceutics, 2023, doi:10.3390/pharmaceutics15061689_

Round 1

Reviewer 1 Report

Why Materials and methods section delayed after results section.

I know results comes after methodology but authors here delayed this section!!

Figures are of good quality.

Figure 6 a and b need to be more clearer and also figure 4

Results are well written as well as discussion section.

Edit some typos

Author Response

REVIEWER 1

Comments and Suggestions, and Answers

We are very grateful for the valuable opinion and comments. All the suggested valuable changes have been done in the corrected version of our paper.

Reviewer comments

1) ,, Why Materials and methods section delayed after results section.

I know results comes after methodology but authors here delayed this section!! "

Answer for Reviewer

We agree with the Reviewer. The Materials and methods section has been moved before the result.

Reviewer comments

2) ,,Figure 6 a and b need to be more clearer and also figure 4"

Answer for Reviewer

Figures have been presented in a more readable way. In addition, clearer figures for individual APIs have been included in the supplements section (Figures S1-S8).

Reviewer 2 Report

The present manuscript reports an attempt of partial comparative analytical evaluation of naproxen and naproxen sodium salt in pure form and selected commercial solid dosage forms from the market. General impression: For the present times the experimental design of the study is obsolete in scientifically irrelevant. There is also almost no industrial applicability of the reported results. For eventual publication, experimental part should be redesigned.

Some detailed suggestions:

1.       Title does not reflect actual content of the manuscript. Main content is comparative evaluation of two forms of naproxen and some commercial pharmaceutical preparations containing naproxen either in acid or sodium salt form.

2.       Line 46 – term ointment should be replaced with dermal preparations – I doubt that naproxen is on the market in form of ointment.

3.       Lines 46-49: I doubt ionized form of naproxen is absorbed faster – usually only nonionised form is absorbed through intestinal or dermal barriers.

4.       Lines 66-72: TGA is not optimal method to distiguish between both forms of naproxen (acid vs. salt form).

5.       Line 100: in the statement the influence of drug particle size onto thermal degradation is totally neglected. Tablet samples were pulverised, so the preparation of the samples for analysis most probably influenced particle size of ingredients including API. It is unrealistic to propose the puriti of incorporated drug by only evaluating the thermal degradation parameters obtained by TGA. Nobody would use such experimental setup in present time.

6.       Lines 130-132: The statement is scientifically wrong.

7.       Page 7-8, Table 3: values for Area should be given in rounded form with only 1 decimal place.

8.       Line 185: what is meant wit “proper decomposition”?

9.       Lines 239-241 – science based explanation for the mentioned thermal effects should be given.

10.   Line 245: authors should recheck the dehydration temperature of lactose monohydrate – it is at much lower temperature

11.   Page 14: authors should in experimental part explain how DFT spectra were obtained.

12.   Chapter 2.2: replace term peak (in IR spectra) with term absorption band or just band.

13.   Chapter 2.4: The use of optical microscopy of samples of powdered tablets is irrelevant. Eliminate this chapter from the manuscript.

14.   Pages 19-20 – what is scientific value of using colorimetric analysis of tested sample – tere is no benefit nor scientific value.

15.   Lies 435.436: correct the sentence and add information on structure of active metabolite of nabumetone.

16.   Lines 509-511: did authors really use all mentioned calibration standards in DTA analysis? Was the apparatus DTA or DSC type?

17.   Conclusions should be corrected taking into account critical view onto the experimental plan and the obtained results.

Text of the manuscript should be checked by native english speaking professional.

Author Response

REVIEWER 2

Comments and Suggestions, and Answers

We are very grateful for the valuable opinion and comments. All the suggested valuable changes have been done in the corrected version of our paper.

Reviewer comments

1) ,,Title does not reflect actual content of the manuscript. Main content is comparative evaluation of two forms of naproxen and some commercial pharmaceutical preparations containing naproxen either in acid or sodium salt form. "

Answer for Reviewer

As the Reviewer suggests, the title of the work has been changed: ,,Application of TGA/c-DTA for Distinguishing Between Two Forms of Naproxen in Pharmaceutical Preparations. "

2) ,,Line 46 – term ointment should be replaced with dermal preparations – I doubt that naproxen is on the market in form of ointment. "

Answer for Reviewer

We agree with the Reviewer, of course naproxen is not in the form of an ointment, but a gel. The sentence has been corrected as suggested by the Reviewer. Lines 45-46:
,, Naproxen comes in tablets, suppositories and dermal preparations for external use. "

3) ,,Lines 46-49: I doubt ionized form of naproxen is absorbed faster – usually only nonionised form is absorbed through intestinal or dermal barriers. "

Answer for Reviewer

A new sentence and literature has been added in this regard. Lines 48-53: ,, Naproxen is a weak acid with pKa=4.15, which determines the rate of its absorption [6, 7]. On the other hand, as it is known, that weak acids in the form of salts dissolve faster in an aqueous environment. Therefore, naproxen in the sodium form is more rapidly dissolved in the environment of body fluids and more rapidly absorbed into the plasma [6-8]. This allows for a faster analgesic effect of naproxen in the form of salt compared to naproxen acid, which has been confirmed in clinical trials [8]."

4) ,,Lines 66-72: TGA is not optimal method to distiguish between both forms of naproxen (acid vs. salt form). "

Answer for Reviewer

We agree with the Reviewer that TGA is not an optimal method to discriminate between forms of naproxen. However, as our research shows, it is possible. We wanted to show that less advanced thermal analyzes can also be useful. In the introduction, we write that there are more advanced methods that are used to identify the API (Line 66-69: ,,In the pharmaceutical industry, many methods are used to identify the type of API [9-12]. Advanced methods such as NMR, HPLC, FTIR, XRPD or Raman spectroscopy are accurate methods, but often the methodology and performance of determinations are time-consuming, complicated and requires expensive equipment [13-17]." At work, we also use a more advanced method - FTIR. The results obtained with the TGA method supported by c-DTA coincide with the FTIR method.

5) ,,Line 100: in the statement the influence of drug particle size onto thermal degradation is totally neglected. Tablet samples were pulverised, so the preparation of the samples for analysis most probably influenced particle size of ingredients including API. It is unrealistic to propose the puriti of incorporated drug by only evaluating the thermal degradation parameters obtained by TGA. Nobody would use such experimental setup in present time. "

Answer for Reviewer

As the Reviewer suggests, the sentence has been reformulated. Information on the impact of particle size on thermal analysis has also been added. Lines 201-206: ,,This indicates high purity of the tested preparations This indicates that the pharmaceutical preparations contain the same API, i.e. naproxen in the form of acid. The tablet mass was pulverized so that the consistency of the tested pharmaceutical preparations was the same as the standard. This is important because the size of the particles has been shown to influence thermal behavior. As the research has shown, the decomposition onset temperature is lowered for smaller particles [38, 39]. "

6) ,,Lines 130-132: The statement is scientifically wrong. "

Answer for Reviewer

Sentence has been deleted. Lines 239-240: ,,The shift in decomposition towards higher temperatures proves a good selection of excipients in the formulation of the finished drug [23, 32]. "

7) Page 7-8, Table 3: values for Area should be given in rounded form with only 1 decimal place.

Answer for Reviewer

As Reviewer indicated, the values for the Area have been rounded to one place after the decimal point (Tables 4 and 7).

8) ,,Line 185: what is meant wit “proper decomposition”? "

Answer for Reviewer

The sentence has been reformulated. Line 285: ,,The TG curve indicates that the main thermal decomposition of NSS occurs at 375.6°C. "

9) ,,Lines 239-241 – science based explanation for the mentioned thermal effects should be given. "

Answer for Reviewer

Additional effects are related to the thermal decomposition of the excipients. For better visualization, additional figures (Figures S1-S8) have been included in the supplementary materials with marked thermal events for API and excipients on the DTG curves.

10)  ,,Line 245: authors should recheck the dehydration temperature of lactose monohydrate – it is at much lower temperature "

Answer for Reviewer

The sentence has been reworded and expanded. Lines 346-359: ,, Another excipient-derived peak in the DTG curve was recorded only for the NS2 and NS4 samples, with a maximum mass loss at 172.3°C and 181°C, respectively. Samples NS2 and NS4 are the only ones that contain lactose monohydrate. Lactose monohydrate has two weight losses related to water loss. As research shows, the first mass loss occurs at a temperature between 130-170°C (with a maximum at 150°C) and is associated with the loss of surface water, the second at a temperature of about 220°C and is associated with the loss of water of crystallization [26, 43, 50, 51, 52]. The recorded peak probably comes from lactose monohydrate. Although the peak values recorded for the NS2 and NS4 samples, are slightly higher than for lactose monohydrate, Altamimi M. J. et al. show that lactose added to pharmaceutical preparations is produced by many manufacturers and is often in the form of anomers (α, and β) [53]. The anomers differ in their physical properties [53, 54]. The authors showed differences in the anomer composition of 19 commercially available lactose monohydrates on the market and showed the need to monitor lactose composition in terms of its anomers in the pharmaceutical industry [53]. "

11) ,,Page 14: authors should in experimental part explain how DFT spectra were obtained. "

Answer for Reviewer

Thanks for your comment. All the descriptions concerning DFT were already provided. Please see, lines: 130-136: ,, 2.4. Density functional theory calculations

The structure of the naproxen acid and sodium salt was constructed in the Avogadro software [34] and further optimized using Orca program package [35] at B3LYP/def2-TZVP level of density functional theory. The frequencies were obtained at the same level of theory and further processed by Multiwfn software [36]. The results extracted from Multiwfn software were further plotted together with experimentally obtained FTIR spectra. "

12) ,,Chapter 2.2: replace term peak (in IR spectra) with term absorption band or just band. "

Answer for Reviewer

As pointed out by the Reviewer in the new Chapter 3.2 the word peak has been changed to band.

13) ,,Chapter 2.4: The use of optical microscopy of samples of powdered tablets is irrelevant. Eliminate this chapter from the manuscript. "

Answer for Reviewer

At the suggestion of the Reviewer, the entire chapter was removed from the work. Lines: 506-531.

14) ,,Pages 19-20 – what is scientific value of using colorimetric analysis of tested sample – tere is no benefit nor scientific value. "

Answer for Reviewer

Colorimetric analyzes are often used to assess the impact of storage conditions on APIs or to assess interactions between API and an excipient (especially in the case of the Millard reaction) [1-3]. In the work, we assessed the suitability of the colorimetric method to assess differences in the color of naproxen acid and sodium salt and pharmaceutical preparations. Unfortunately, studies have shown that this method is not sensitive enough to distinguish between the two forms of naproxen. However, it allowed to demonstrate that the similarity of drug formulations to naproxen standards, which may partially indicate the appropriate API in the formulation. However, as we write, this is not a very specific method in this case and one should not make a judgment only on its basis.

[1] Ramos P., Broncel M. Influence of storage conditions on the stability of gum arabic and tragacanth. Molecules 2022, Vol.27, No.5, p.1-16, id. art. 1510

[2] 37. Echavarria A.P., Pagan J., Ibarz A. Kinetics of color development in glucose/Amino Acid model systems at different temperatures. Sci. Agropec. 2016, 7(1), p.15-21.

[3] Ramos P. Compatibility studies of selected mucolytic drugs with excipients used in solid dosage forms: thermogravimetry analysis. Farmacia 2021, Vol.69, No.3, p.585-594

15) ,,Lies 435.436: correct the sentence and add information on structure of active metabolite of nabumetone. "

Answer for Reviewer

At the suggestion of the Reviewer, the new information and Figure has been added. Lines 568-572: ,, (6-methoxy-2-naphthyl)acetic acid is a lower homolog of naproxen showing strong inhibition of cyclooxygenase mainly isoform 2 (COX-2) [2, 7].

Figure 17. Figure 15. Chemical structure of (a) nabumetone, and (b) active metabolite of nabumetone (6-methoxy-2-naphthyl)acetic acid [5, 7]. "

16) ,,Lines 509-511: did authors really use all mentioned calibration standards in DTA analysis? Was the apparatus DTA or DSC type? "

Answer for Reviewer

The apparatus was of the DTA type. All standards were used in the study (the photo shows the standards used in the study).

Figure. high-purity reference materials (In, Sn, Zn, Al, BaCO3, and Au).

17) ,,Conclusions should be corrected taking into account critical view onto the experimental plan and the obtained results. "

Answer for Reviewer

As suggested by the reviewer, the conclusions have been redrafted. Lines 617-635:

,,The research has shown that thermogravimetry (TGA) assisted by calculated differential thermal analysis (c-DTA), helps distinguish between two forms of naproxen - acid and sodium salts. This allows for the initial identification of the type of naproxen form in commercial pharmaceutical preparations.

TGA, in combination with c-DTA allows to distinguish naproxen in the form of acid and sodium salt by registering specific thermal events for each form, which was not possible with some pharmacopoeial methods such as UV-Vis or HPLC. For some pharmaceutical preparations, it was also possible to record thermal events from excipients.

In addition, using a close structural analog of naproxen (nabumetone), high specificity for the evaluated compounds of the applied thermal methods was demonstrated.

Research has shown that TGA supported by c-DTA is as effective in determining the type of naproxen form in various pharmaceutical preparations as the commonly used FTIR technique.

However, it should be remembered that despite many advantages such as speed and simplicity of measurement, small sample volume, easy preparation, repeatability or low cost of measurement, the applied thermal methods also have some limitations, e.g. in order to obtain high repeatability, the measurements must be made under the same conditions, using the same measuring crucible, sample weight and often using the same apparatus. "

Reviewer 3 Report

This article reports that application of the thermal analyses (DTA and c-DTA) to distinguish the form of naproxen in the pharmaceutical tablets. The results were carefully obtained and discussed with comparison to the other analytical methods such as FT-IR, HPLC, and UV spectra. The knowledge of them is worthy for publication of Pharmaceutics. However, some points should be revised before publication. The comments are listed here and please consider them to improve the manuscript.

1.      The order of “materials and methods” and “results and discussion” may not be correct according to the template of Pharmaceutics.

2.      The authors mentioned that the thermal analyses performed in this study are superior to the other methods from view points of time, cost, and complexity (page 2, line 59-65). Hence, could you show the readers the comparison of them in discussions or conclusions.

3.      The referred figure number may not be correct? In page 4, line 116-117, DTG curve (Figure 2a) and TG curve (Figure 1) are written. However, Figure 1 is the chemical structures, also Figure 2a is not shown (Figure 2 is the TGA curves).

4.      In Figure 5, the different of the TG curves was explained from the drug content in the formulations. (page 8, line 189-191). Please show both weight of tablet and API in Table 7 for easy comparison among the formulations and the those of NA and NS.

5.      It is known that sodium naproxen can form hydrate forms (mono-, di-, and tetra-hydrates) in addition to anhydrate form (https://doi.org/10.1007/s11095-012-0872-8). The difference the DTA curves among the NS formulations might reflect these pseudo-polymorphs. Please add further discussion there corresponding to the weight loss pattern and the form of the hydrate (page 9, line 212-224).

6.      In Table 5, the NS formulations show the different decomposition stages. NS2 gave only the stage VII with higher temperature. What causes this difference among the formulations. The authors mentioned the decomposition points of the excipients (page 10, lie 242-256). Should they affect the thermal profiles?

7.      In Figures 8 and 9, the legends were represented as the commercial names. Please use legends with the other Figures (NA1, NA2, …).

8.      The UV peaks were represented as peaks 3 and 4 (page 16, line 357), but the peak with wavenumber (e.g. the peak at 316 nm) is better for understanding.

9.      In Figures 11 and 12, the authors mentioned that the retention times of NA and NS was slight different. However, NS3 gave it at 4.23 min, which closes to that of NA1-3. Can the NA and NS be distinguished from HPLC chromatogram?

That’s all.

Author Response

REVIEWER 3

Comments and Suggestions, and Answers

We are very grateful for the valuable opinion and comments. All the suggested valuable changes have been done in the corrected version of our paper.

Reviewer comments

1) ,,The order of "materials and methods" and "results and discussion" may not be correct according to the template of Pharmaceutics. "

Answer for Reviewer

We agree with the Reviewer. The Materials and methods section has been moved before the result.

2) ,,The authors mentioned that the thermal analyses performed in this study are superior to the other methods from view points of time, cost, and complexity (page 2, line 59-65). Hence, could you show the readers the comparison of them in discussions or conclusions. "

Answer for Reviewer

We agree with the Reviewer. The literature in this regard has been expanded and this has been underlined in the revised conclusions.

Lines 77-78: ,,…and quality control in pharmacy [29-31]. "

  1. Harmathy, Z.; Konkoly Thege, I. Application of modern thermal methods in pharmaceutical analysis. Acta Pharm. Hung. 1994, 64(1): 9-15.
  2. Giron, D. Applications of thermal analysis and coupled techniques in pharmaceutical industry. J. Therm. Anal. Calorim. 2002, 68: 335-357.
  3. Monajjemzadeh, F.; Ghaderi, F. Thermal Analysis Methods in Pharmaceutical Quality Control. J. Mol. Pharm. Org. Process Res. 2015, 3: 1-2.

Lines 617-635 Conclusions:

,,The research has shown that thermogravimetry (TGA) assisted by calculated differential thermal analysis (c-DTA), helps distinguish between two forms of naproxen - acid and sodium salts. This allows for the initial identification of the type of naproxen form in commercial pharmaceutical preparations.

TGA, in combination with c-DTA allows to distinguish naproxen in the form of acid and sodium salt by registering specific thermal events for each form, which was not possible with some pharmacopoeial methods such as UV-Vis or HPLC. For some pharmaceutical preparations, it was also possible to record thermal events from excipients.

In addition, using a close structural analog of naproxen (nabumetone), high specificity for the evaluated compounds of the applied thermal methods was demonstrated.

Research has shown that TGA supported by c-DTA is as effective in determining the type of naproxen form in various pharmaceutical preparations as the commonly used FTIR technique.

However, it should be remembered that despite many advantages such as speed and simplicity of measurement, small sample volume, easy preparation, repeatability or low cost of measurement, the applied thermal methods also have some limitations, e.g. in order to obtain high repeatability, the measurements must be made under the same conditions, using the same measuring crucible, sample weight and often using the same apparatus. "

3) ,,The referred figure number may not be correct? In page 4, line 116-117, DTG curve (Figure 2a) and TG curve (Figure 1) are written. However, Figure 1 is the chemical structures, also Figure 2a is not shown (Figure 2 is the TGA curves)."

Answer for Reviewer

We agree with the Reviewer. The error has been corrected. Figures 2a and 1 were cited incorrectly instead of 3a and 2. Lines 219-220: ,,NAS has two decomposition stages on the DTG curve (Figures 3a, S1) corresponding with the TG curve (Figures 2, S1). The major mass loss occurs in the first stage with a …".

4) ,,In Figure 5, the different of the TG curves was explained from the drug content in the formulations. (page 8, line 189-191). Please show both weight of tablet and API in Table 7 for easy comparison among the formulations and the those of NA and NS. "

Answer for Reviewer

We agree with the Reviewer. In the new Table 1, the tablet weights of individual pharmaceutical preparations have been added.

5) ,,It is known that sodium naproxen can form hydrate forms (mono-, di-, and tetra-hydrates) in addition to anhydrate form (https://doi.org/10.1007/s11095-012-0872-8). The difference the DTA curves among the NS formulations might reflect these pseudo-polymorphs. Please add further discussion there corresponding to the weight loss pattern and the form of the hydrate (page 9, line 212-224). "

Answer for Reviewer

As the Reviewer points out, the discussion regarding hydrate forms of naproxen sodium salt has been added. Lines 319-326: ,, As is known, naproxen sodium can be in the form of an anhydrate, a monohydrate, two dihydrate polymorphs, and a tetrahydrate [48]. The formation of pseudo-polymorphic forms may affect the recording of thermal curves. Rajada D et al. showed in their studies that anhydrous sodium naproxen is converted to one of the dihydrate polymorphs al-ready at 25°C. On the other hand, at a higher temperature (50°C), anhydrous sodium naproxen is gradually converted into a monohydrate and then into another dihydrate polymorph. The dihydrate polymorphs can transform into tetrahydrate and monohydrate [48]."

6) ,,In Table 5, the NS formulations show the different decomposition stages. NS2 gave only the stage VII with higher temperature. What causes this difference among the formulations. The authors mentioned the decomposition points of the excipients (page 10, lie 242-256). Should they affect the thermal profiles? "

Answer for Reviewer

We agree with the Reviewer. Formulation NS2 in step VII has the same temperature (458.9°C) as the other formulations (NS1, NS3-NS5) in step VI (457°C-466°C). Thermal events on DTG curves are numbered in the order in which they occur, from the lowest to the highest temperature. The NS2 formulation has an additional event (Stage III) occurring at a lower temperature (172.3°C), probably related to the presence of lactose compared to the other formulations. This means that the NS2 preparation has VII stages of decomposition and not VI like the rest of the preparations. But this is normal because NS2 has the highest amount of excipients compared to the other formulations (Table 1). To better clarify this issue, Figures S4-S8 have been added in additional materials to illustrate the thermal decompositions better.

7) ,,In Figures 8 and 9, the legends were represented as the commercial names. Please use legends with the other Figures (NA1, NA2, …)."

Answer for Reviewer

As indicated by the Reviewer, this has been changed in the new Figures 8 and 9. Lines 441 and 459.

8) ,,The UV peaks were represented as peaks 3 and 4 (page 16, line 357), but the peak with wavenumber (e.g. the peak at 316 nm) is better for understanding. "

Answer for Reviewer

We agree with the Reviewer, sentence has been corrected. Line 474: ,,…a peak at 316 nm and peak at 331 nm by 1 nm towards a higher wavelength can be observed. "

9) ,,In Figures 11 and 12, the authors mentioned that the retention times of NA and NS was slight different. However, NS3 gave it at 4.23 min, which closes to that of NA1-3. Can the NA and NS be distinguished from HPLC chromatogram? "

Answer for Reviewer

Thanks for your comment. It is impossible to distinguish NA from NS by the HPLC method because the form which naproxen will take in water depends on the solution's (e.g. mobile phase) pH.

Reviewer 4 Report

The manuscript entitled “Application of TGA and c-DTA to Identify Naproxen Forms in Commercial Pharmaceutical Preparations” investigated different naproxen forms in marketed formulations using analytical tools. The study design is organized and the the results supported the study findings. I have following points, which needs to be addressed

1.      Authors would require rephrasing “Thanks to this, they can be an alternative to more complicated analytical methods that require more 29 time, are expensive and have more complicated sample preparation” to more scientific description rather than the casual statement.

2.      It has been already proven that preformulation tools such as TGA; c-DTA;  FTIR; optical microscope techniques can be used to determine the physical forms of APIs, what authors need to address the novelty of the manuscript

3.      Was there any effect of different naproxen forms in marketed formulations on the bioavailability/stability? If so, quote the literature in introduction.

Author Response

REVIEWER 4

Comments and Suggestions, and Answers

We are very grateful for the valuable opinion and comments. All the suggested valuable changes have been done in the corrected version of our paper.

Reviewer comments

1) ,, Authors would require rephrasing “Thanks to this, they can be an alternative to more complicated analytical methods that require more 29 time, are expensive and have more complicated sample preparation” to more scientific description rather than the casual statement. "

Answer for Reviewer

We agree with the Reviewer, sentence has been corrected. Lines 28-29: ,,This indicates the potential possibility of using TGA supported by c-DTA as an alternative method."

2) ,,It has been already proven that preformulation tools such as TGA; c-DTA;  FTIR; optical microscope techniques can be used to determine the physical forms of APIs, what authors need to address the novelty of the manuscript"

Answer for Reviewer

As indicated in the scientific literature and pharmacopoeia guidelines, thermal methods such as TGA or DTA are mainly used to: determine the thermal decomposition profile of API, assess the polymorphism of the active substance, assess the relative humidity content in drugs, or preliminary assessment of API-excipient compatibility. In the scientific literature, we have not found any works that would concern the possibility of assessing the type of API (including the form of naproxen) in pharmaceutical preparations. In our earlier work, we demonstrated the possibility of using the TGA and c-DTA methods to assess theophylline and aminophylline content in pharmaceutical preparations (Ramos P. Application of Thermal Analysis to Evaluate Pharmaceutical Preparations Containing Theophylline. Pharmaceuticals 2022, 15(10), 1268: 1-19. It seems to us that there is a need to look for simpler, alternative methods of assessing API content in drugs. It seems to us that due to the numerous advantages, thermal methods can be an alternative here.

3) ,,Was there any effect of different naproxen forms in marketed formulations on the bioavailability/stability? If so, quote the literature in introduction. "

Answer for Reviewer

New text and citations have been added in the Introduction section. Lines 48-53: ,,Naproxen is a weak acid with pKa=4.15, which determines the rate of its absorption [6, 7]. On the other hand, as it is known, that weak acids in the form of salts dissolve faster in an aqueous environment. Therefore, naproxen in the sodium form is more rapidly dissolved in the environment of body fluids and more rapidly absorbed into the plasma [6-8]. This allows for a faster analgesic effect of naproxen in the form of salt compared to naproxen acid, which has been confirmed in clinical trials [8]. "

Reviewer 5 Report

Summary

The manuscript entitled ‘Application of TGA and c-DTA to Identify Naproxen Forms in Commercial Pharmaceutical Preparations’ presents thermoanalytical investigation of the acid or sodium salt form of naproxen in different formulations.

The authors examined how the two presented forms can be distinguish according to their thermal behavior. Although the manuscript contain interesting results the novelty of the work is not highlighted enough. In addition, and not all conclusions are supported with sufficient research results.

For all the above-mentioned reasons, I would recommend the acceptance of this publication with major revision after consideration of some comments and remarks addressed.

General comments

1. It is a bit confusing that the authors present first the results and only after that the materials and methods. Please change the order of chapter 2 and 3. First describe the ‘Materials and methods’ and then the ‘Results and discussion’.

2. Please use the introduced abbreviations consistently throughout the entire manuscript. E.g. on page 4, in Line 116 the authors write ‘pure naproxen acid’. The previously mentioned ‘NAS’ abbreviation could be used here.

3. Page 3, Line 96: Please correct the word ‘congaing’ to ‘containing’.

4. Page 4, Line 116: The authors mention Figure2a but there is no Figure2a only Figure2.

5. Page 4, Line 124: The authors mention Figure2b but there is no Figure2b only Figure2.

6. Figure 8 and Figure 9: Please use the same name and same color of the samples as it was used in the previous Figures.

7. Page 15, Line 333: Correct Fig.2b to Figure1a.

8. Page 26, Line 560: What does the ‘VAN’ abbreviation mean?

Comments

1. Page 9, Line 214: ‘Naproxen sodium salt undergoes four-step decomposition.’ It would be easier to interpret if these steps would be indicated in Figure6.

2. Page 10, Line 250: ‘…probably from microcrystalline cellulose…’ It would help the explanation if the references (data of the excipients) were also shown in the Figures.

3. Figure 11 and 12: HPLC results of the pure APIs do not shown. It would be better if reference samples can be seen.

4. Page 17, Line 389-192: ‘When analyzing the image of tablets containing naproxen in the form of acid and salt as APIs, the tablet's homogeneous, compact cross-sectional structure can be seen. The good mixing of the formulation ingredients and the use of appropriate punch pressure during tableting confirms this’

I would not say based on these images that these are homogeneous. A chemical mapping could determine the homogeneity, or dissolution tests or CU measurements could give information about the homogeneity. According to the optical microscopic images what can we see? Size of the particles, shape of the particles, but not the composition and the homogeneity…

How can the mixing and the punch pressure confirm this statement? It is not clear for me.

I am not sure that these optical microscopic images give relevant information from the point of view of the research.

Minor editing of English language required.

Author Response

REVIEWER 5

Comments and Suggestions, and Answers

We are very grateful for the valuable opinion and comments. All the suggested valuable changes have been done in the corrected version of our paper.

Reviewer comments (General comments)

1) ,,It is a bit confusing that the authors present first the results and only after that the materials and methods. Please change the order of chapter 2 and 3. First describe the ‘Materials and methods’ and then the ‘Results and discussion’. "

Answer for Reviewer

We agree with the Reviewer. The Materials and methods section has been moved before the result.

2) ,,Please use the introduced abbreviations consistently throughout the entire manuscript. E.g. on page 4, in Line 116 the authors write ‘pure naproxen acid’. The previously mentioned ‘NAS’ abbreviation could be used here. "

Answer for Reviewer

We agree with the Reviewer, sentence has been corrected. Lines 219-220: ,, NAS has two decomposition stages on the DTG curve (Figures 3a, S1) corresponding with the TG curve (Figures 2, S1). "

3) ,,Page 3, Line 96: Please correct the word ‘congaing’ to ‘containing’. "

Answer for Reviewer

The word has been corrected. Line 197: ,, Thermogravimetry curves registered for all tested pharmaceutical preparations containing naproxen acid (NA1-NA4) have the same course and shape as a pattern of NAS (Figure 2, Table 2).’’

4) ,,Page 4, Line 116: The authors mention Figure2a but there is no Figure2a only Figure2. "

Answer for Reviewer

We agree with the Reviewer. The error has been corrected. Figures 2a was cited incorrectly instead of 3a. Lines 219-220: ,,NAS has two decomposition stages on the DTG curve (Figures 3a, S1) corresponding with the TG curve (Figures 2, S1). The major mass loss occurs in the first stage with a …".

5) ,,Page 4, Line 124: The authors mention Figure2b but there is no Figure2b only Figure2. "

Answer for Reviewer

We agree with the Reviewer. The error has been corrected. Figures 2b was cited incorrectly instead of 3b. Lines 231: ,,…that most of the compound is degraded in the first stage (Figure 3b, Table 3) [19, 40, 41]. ".

6) ,,Figure 8 and Figure 9: Please use the same name and same color of the samples as it was used in the previous Figures. "

Answer for Reviewer

As indicated by the Reviewer, this has been changed in the new Figures 8 and 9. Lines 441 and 459.

7) ,,Page 15, Line 333: Correct Fig.2b to Figure1a. "

Answer for Reviewer

As indicated by the Reviewer, this has been changed. Lines 451-452: ‘’On the other hand, the spectra of acid naproxen (Fig. 9a) show several band, at 1720 cm-1 is present the signal ascribed to C=O from carboxylic acids. "

8) ,,Page 26, Line 560: What does the ‘VAN’ abbreviation mean? "

Answer for Reviewer

The HPLC methodology section has been redrafted. Lines 148-158: ,, 2.6. HPLC methodology

The high-performance liquid chromatography (HPLC) apparatus (UltiMate 3000, Thermo Fisher Scientific, Czech Republic) with UV-VIS (VWD-3100) detector was used to separate the naproxen acid and naproxen sodium salt. The flow rate was adjusted to 1200 μL/min. The mobile phases consisted of mixtures of 40% acetonitrile and 60 % 0.01 M orthophosphoric acid water solution. The separation of naproxen sodium salt and naproxen acid were conducted in 40°C with HPLC column type C18 (150 mm × 4.6 mm, 2.6 μm; Phenomenex, Czech Republic). The injection of the volume was 20 μL. The retention time for naproxen acid were obtained in 4.70 min and for naproxen sodium salt in 4.0 min. The wavelength or measurement was 230 nm and the limit of quantification was 0.5 mg/L for both (naproxen acid and naproxen sodium salt). "

Reviewer comments (Comments)

1) ,,Page 9, Line 214: ‘Naproxen sodium salt undergoes four-step decomposition.’ It would be easier to interpret if these steps would be indicated in Figure6. "

Answer for Reviewer

Figures have been presented in a more readable way. In addition, clearer figures for individual APIs have been included in the supplements section (Figures S1-S8).

2) ,,Page 10, Line 250: ‘…probably from microcrystalline cellulose…’ It would help the explanation if the references (data of the excipients) were also shown in the Figures. "

Answer for Reviewer

Figures have been presented in a more readable way. In addition, clearer figures for individual APIs have been included in the supplements section (Figures S1-S8).

3) ,,Figure 11 and 12: HPLC results of the pure APIs do not shown. It would be better if reference samples can be seen. "

Answer for Reviewer

Thanks for your comment. The pure APIs were used as HPLC standards only, as noted in the manuscript.

4) ,,Page 17, Line 389-192: ‘When analyzing the image of tablets containing naproxen in the form of acid and salt as APIs, the tablet's homogeneous, compact cross-sectional structure can be seen. The good mixing of the formulation ingredients and the use of appropriate punch pressure during tableting confirms this’

I would not say based on these images that these are homogeneous. A chemical mapping could determine the homogeneity, or dissolution tests or CU measurements could give information about the homogeneity. According to the optical microscopic images what can we see? Size of the particles, shape of the particles, but not the composition and the homogeneity…

How can the mixing and the punch pressure confirm this statement? It is not clear for me.

I am not sure that these optical microscopic images give relevant information from the point of view of the research. "

Answer for Reviewer

At the suggestion of the Reviewer, the entire chapter was removed from the work. Lines: 506-531.

Round 2

Reviewer 5 Report

The reviewer thanks the authors for considering the issues raised in the first round of the review. The manuscript has improved a lot and is ready for publication.

Author Response

REVIEWER 5

Comments and Suggestions, and Answers

We are very grateful for Your valuable opinion, comments, and manuscript acceptance.
